# TAF4, a subunit of transcription factor II D, directs promoter occupancy of nuclear receptor HNF4A during post-natal hepatocyte differentiation

Daniil Alpern[1], Diana Langer[1], Benoit Ballester[2,3], Stephanie Le Gras[1], Christophe Romier[4], Gabrielle Mengus[1], Irwin Davidson[1]*

[1]Department of Functional Genomics and Cancer, Institut de Genetique et de Biologie Moleculaire et Cellulaire, CNRS/INSERM/UDS, Illkirch, France; [2]Laboratoire TAGC, Aix-Marseille Université, UMR1090, Marseille, France; [3]Laboratoire TAGC, INSERM, UMR1090, Marseille, France; [4]Department of Integrated Structural Biology, Institut de Genetique et de Biologie Moleculaire et Cellulaire, CNRS/INSERM/UDS, Illkirch, France

**Abstract** The functions of the TAF subunits of mammalian TFIID in physiological processes remain poorly characterised. In this study, we describe a novel function of TAFs in directing genomic occupancy of a transcriptional activator. Using liver-specific inactivation in mice, we show that the TAF4 subunit of TFIID is required for post-natal hepatocyte maturation. TAF4 promotes pre-initiation complex (PIC) formation at post-natal expressed liver function genes and down-regulates a subset of embryonic expressed genes by increased RNA polymerase II pausing. The TAF4–TAF12 heterodimer interacts directly with HNF4A and in vivo TAF4 is necessary to maintain HNF4A-directed embryonic gene expression at post-natal stages and promotes HNF4A occupancy of functional cis-regulatory elements adjacent to the transcription start sites of post-natal expressed genes. Stable HNF4A occupancy of these regulatory elements requires TAF4-dependent PIC formation highlighting that these are mutually dependent events. Local promoter-proximal HNF4A–TFIID interactions therefore act as instructive signals for post-natal hepatocyte differentiation.

*For correspondence: irwin@igbmc.fr

**Reviewing editor**: Michael R Green, Howard Hughes Medical Institute, University of Massachusetts Medical School, United States

## Introduction

TFIID plays a critical role in RNA polymerase II (Pol II) pre-initiation complex (PIC) formation. The TATA-box binding protein (TBP) and 13–14 TBP-associated factors (TAFs) assemble to form TFIID from an association between the 'core complex' comprising the TAF4–TAF12 and TAF6–TAF9 histone-fold heterodimers together with TAF5 and a second module comprising TBP, TAF1, TAF2, and TAF7 (*Cler et al., 2009*; *Bieniossek et al., 2013*). The TAF10–TAF8 dimer associates with the core complex, while the role of the TAF10–TAF3 and TAF11–TAF13 dimers is less well described (*Bieniossek et al., 2013*).

Extensive genetic analysis of TAFs has been performed in yeast (*Shen et al., 2003*), but their function in mammalian cells particularly in complex physiological processes remains poorly characterised. The best understood are those of the cell-specific TAF paralogues like TAF7L that plays a critical role in male germ cell development and in adipocytes (*Cheng et al., 2007*; *Zhou et al., 2013*, *2014*) or TAF4B essential for male and female fertility (*Falender et al., 2005*; *Voronina et al., 2007*). In contrast, much less is known about the ubiquitously expressed 'core' TAFs. For example, mice carrying loss of function alleles for TAF10, TAF8, TAF7, or TBP die between blastocyst and pre-implantation stages upon depletion of the maternal protein (*Voss et al., 2000*; *Martianov et al., 2002*; *Mohan et al., 2003*;

**eLife digest** To decode the information contained within a gene, a number of processes need to occur. For example, the DNA sequence that makes up the gene needs to be copied to make a molecule of RNA, which is then translated to build the corresponding protein. The first steps in the manufacture of RNA involve a structure called a 'pre-initiation complex' moving an enzyme called RNA polymerase II to the start of the gene that needs to be copied.

The pre-initiation complex is made up of many types of protein, including a set of proteins called TAFs. However, the way that these proteins work in mammals is not well understood. There are good reasons for this: proteins are often studied by seeing what happens when the protein is removed, but many TAFs are so important that removing them is lethal.

Alpern et al. have now studied the function of TAF4 by removing this protein from mouse liver cells. This causes severe hypoglycemia (that is, a drop in sugar levels in the blood). Moreover, it seems as if these cells start dying before they become fully mature. In liver cells lacking TAF4, some 1408 genes that are normally turned on just after birth are not properly switched on; these genes are necessary for the metabolic functions of the liver. Furthermore, 776 genes that are normally turned off after birth continue to be expressed. It seems that the absence of TAF4 sometimes disrupts the formation of the pre-initiation complex, which would slow down the production of RNA. However, it can also have the opposite effect by increasing the activity of RNA polymerase II, hence making too many copies of RNA from some genes.

Alpern et al. also find that TAF4 is needed to allow a protein called HNF4A, which is important in the development of the liver and in controlling metabolism, to interact with over 7000 important DNA sequences. Mutations in HNF4A are responsible for a syndrome known as Maturity Onset of Diabetes in the Young. The next stage in this work will be to explore if these mutations influence the interaction between HNF4A and TAF4, and if they do, whether these changes contribute to this form of diabetes.

*Gegonne et al., 2012*). TAF4 and TAF10 have also been analysed in somatic tissues where they are necessary for normal development of the mouse epidermis (*Indra et al., 2005*; *Fadloun et al., 2007*). In contrast, in adult epidermis, loss of TAF10 has no evident phenotype whereas TAF4 plays a critical role in keratinocyte proliferation acting as a cell autonomous and non-cell autonomous tumour suppressor (*Fadloun et al., 2007*).

Despite these studies, the role of TAFs in physiological processes in vivo remains largely unknown and the molecular basis of their action is poorly characterised. In this study, we show that TAF4 is essential for activation of the post-natal hepatocyte gene expression programme. The TAF4–TAF12 heterodimer interacts with the nuclear receptor HNF4A to promote its binding to conserved and functional cis-regulatory elements located close to the transcription start sites of liver function genes. The stable binding of HNF4A to these elements is thus dependent on concomitant pre-initiation complex formation. These results reveal that HNF4A interaction with the basal transcription machinery controls not only PIC formation but also the occupation of its cognate regulatory elements.

## Results

### TAF4 inactivation in hepatocytes leads to cholestatic lesions and early post-natal death associated with hypoglycemia

Immunofluorescence revealed TAF4 expression in P12 wild-type (WT) hepatocytes, mesenchymal tissues, endothelial and smooth muscle cells around the hepatic veins and arteries, and cholangiocytes lining the bile ducts (*Figure 1A*). TAF4 and TBP are expressed in late embryonic hepatocytes and TBP remains strongly expressed in adult hepatocytes, while TAF4 expression is reduced (*Figure 1—figure supplement 1*). TAF4 and TBP are therefore expressed from the embryonic to adult stages, with high expression in neonatal liver. Reduced TAF4 expression in adult hepatocytes has been previously reported (*D'Alessio et al., 2011*), but we do not observe the reported strong loss of TBP expression.

Mice carrying floxed alleles of the *Taf4a* gene were crossed with animals expressing Cre recombinase under the control of the albumin (Alb) promoter to inactivate *Taf4* in post-partum murine hepatocytes

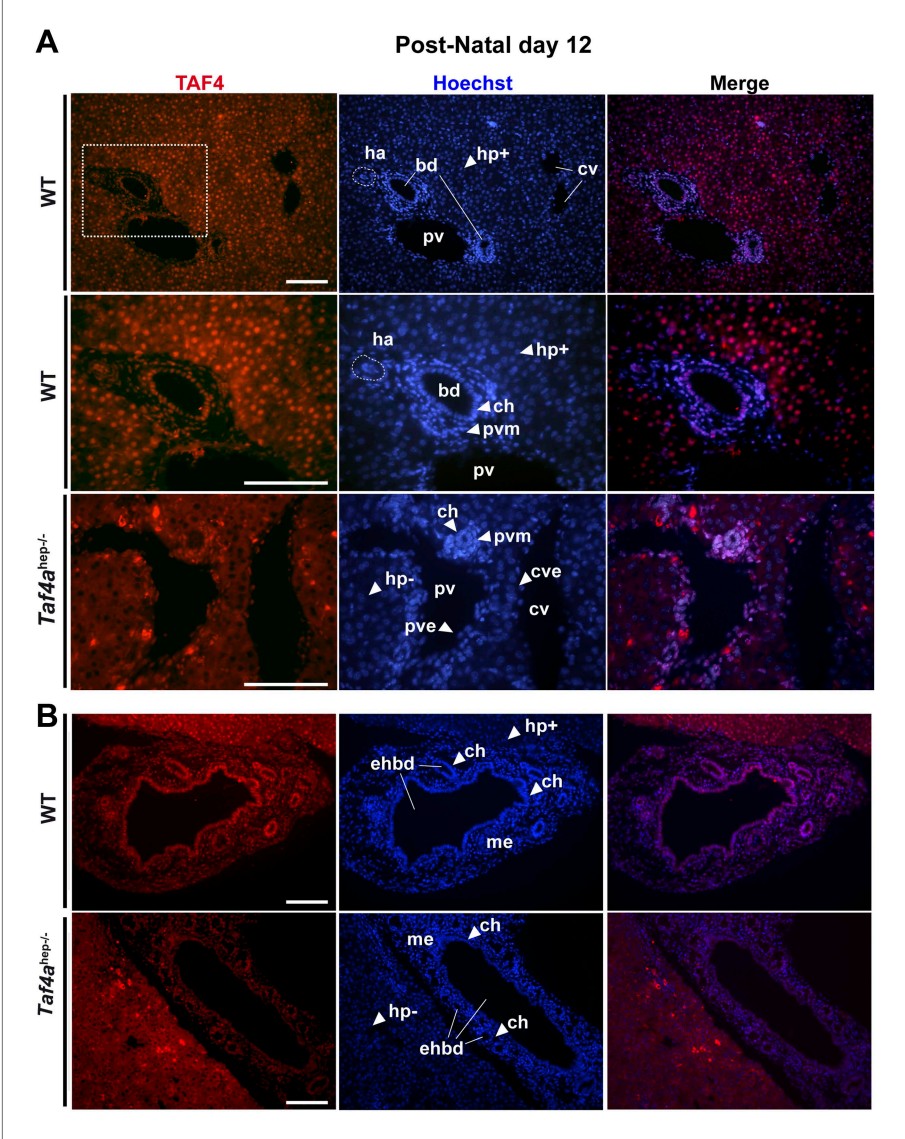

**Figure 1**. Expression of TAF4 in neonatal liver. (**A**) The first two panels show immunostaining for TAF4 in sections of WT liver at P12, while the third panel shows immunostaining in sections from TAF4 mutant liver. The boxed region in the upper panel is blown up in the lower panel. Bd; bile duct, ha; hepatic artery, pv; portal vein, and cv; central vein, Hp+; TAF4-expressing hepatocyte, Hp−; TAF4 negative hepatocyte after Cre-mediated inactivation, ch; cholangiocyte, pvm; portal vein mesenchyme, cve; central vein endothelium. (**B**) Immunofluorescence for TAF4 expression in sections through WT and *Taf4a*hep−/− liver illustrating persistent expression in the extra-hepatic bile duct (ehbd) cholangiocytes and associated mesenchyme (scale bar = 100 µm).

The following figure supplement is available for figure 1:

**Figure supplement 1**. Comparison of TBP and TAF4 expression in late embryonic to adult stage liver.

(*Taf4aaf4*hep−/−). TAF4 expression is lost from hepatocytes by P12 (*Figure 1A–B*), although it remains in the portal vein mesenchyme, endothelial cells, cholangiocytes of extra-hepatic bile ducts, and bile ducts close to the liver hilum, but not in cholangiocytes forming peripheral bile ducts (*Figure 1A,B*).

   *Taf4a*hep−/− animals are born at Mendelian ratios and no prenatal abnormalities or lethality were observed (data not shown). *Taf4a*hep−/− animals are of normal weight and size at birth, show retarded growth by P12, begin to die between P12 and P14, and all are dead by P21 (*Figure 2A,B*). Mutants display severe physiological and morphological alterations caused by perturbed liver function. Cholestatic

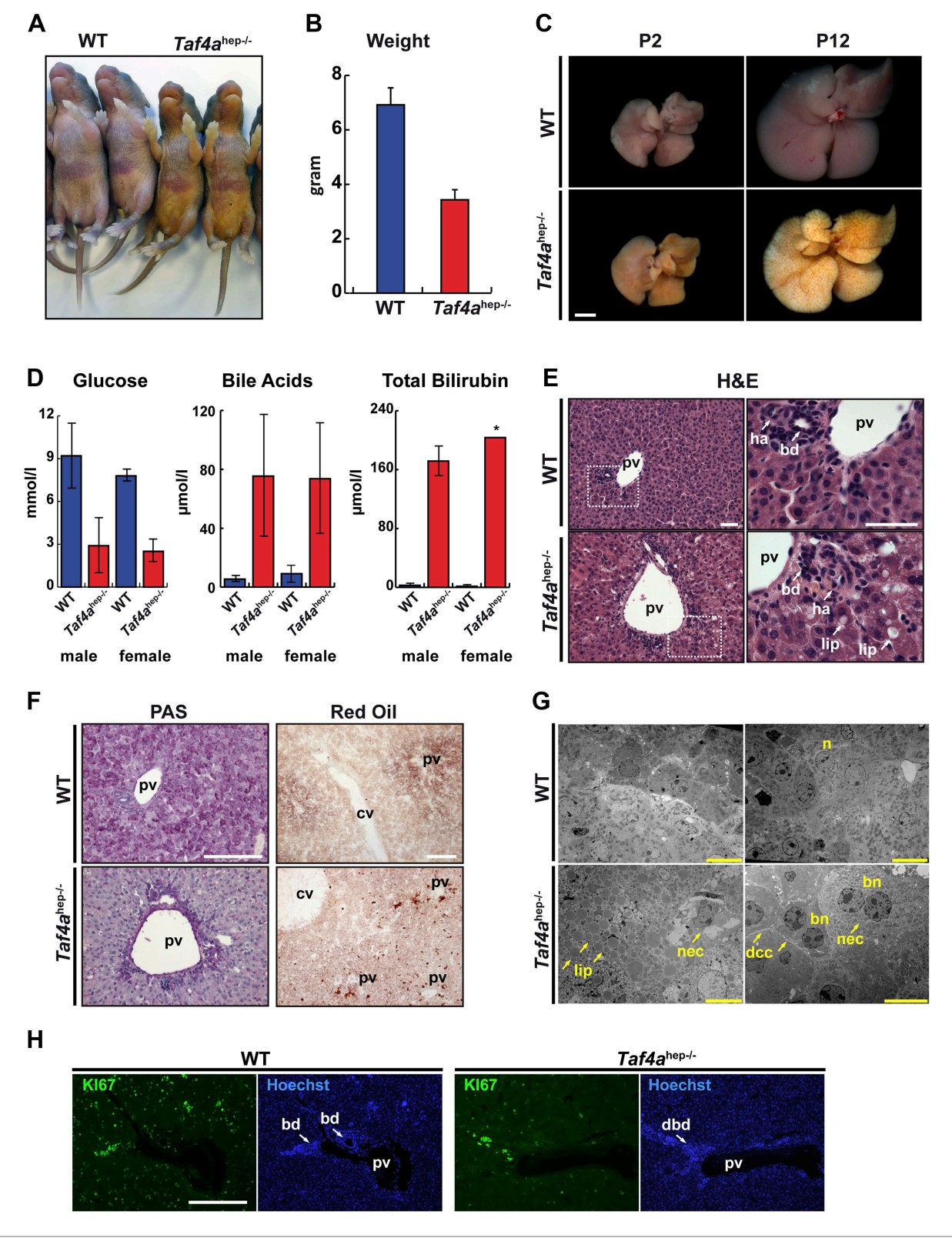

**Figure 2**. Physiopathology of TAF4 inactivation in post-natal hepatocytes. (**A**) WT and *Taf4a*[hep−/−] animals at P4. (**B**) Body weight of WT and *Taf4a*[hep−/−] animals at P12, N = 20. (**C**) Livers from animals at P2 and 12 (scale bar = 2 mm). (**D**) Analysis of serum from WT and *Taf4a*[hep−/−] animals for the indicated parameters at P10, N = 6. (**E**) Hematoxylin–eosin staining of liver sections. Right panel shows a higher resolution of the boxed area of the left panel
*Figure 2. Continued on next page*

*Figure 2. Continued*

(scale bar = 30 μm). Pv; portal vein, bd; bile duct, dbd; defective bile duct; ha; hepatic artery, lip; lipid droplets. (**F**) Periodic acid–Schiff and oil red O staining of livers (scale bar = 100 μm). Cv; central vein. (**G**) Electron micrographs of livers (scale bar = 10 μm). Nec; necrosis, n; nucleus, bn; bi-nucleate cell, dcc; defective cell–cell-contacts.

The following figure supplement is available for figure 2:

**Figure supplement 1**. Physiopathology of TAF4 inactivation in post-natal hepatocytes.

lesions of mutant livers were evident at P2 with marked jaundice of the animals whose severity rapidly increases over the next days (*Figure 2A,C*). By P12, knockout animals display 50% reduced body weight and severe jaundice (*Figure 2A,B*) and their livers become completely yellow with visible patches of bile deposits within the parenchyma (*Figure 2C*).

Blood plasma analysis revealed severe hypoglycemia in *Taf4a*[hep−/−] mice with glucose levels below 3.0 mmol/l in both males and females that is the most probable cause of early lethality (*Figure 2D*). In accordance with their jaundiced appearance, we also detected high levels of bile acids and bilirubin in the plasma of the *Taf4a*[hep−/−] mice (*Figure 2D*). In contrast, the levels of free fatty acids (FFA), triglycerides (TA), and total cholesterol do not significantly change, but HDL cholesterol is considerably reduced compared to WT mice (*Figure 2—figure supplement 1*). TAF4 knockout in post-natal hepatocytes therefore causes severe liver malfunction, hypoglycemia, and dysfunctional bile and lipid transport and metabolism.

TAF4 mutant livers display alterations in parenchyma and lobule organisation with disorganised hepatocyte disposition in lobules and non-aligned hepatic plates due to variations in cell size (*Figure 2E*). Periportal hepatocytes are markedly vacuolated and often display foamy intra-cytoplasmic inclusions indicating steatosis (white arrows in *Figure 2E*, oil red staining in *Figure 2F–G*). PAS staining revealed a lack of glycogen in TAF4 knockouts (*Figure 2F*), which along with hypoglycemia shows defective carbohydrate metabolism in *Tafa4*[hep−/−] mice. We also observed large and bi-nucleate hepatocytes mostly in pericentral areas, many necrotic cells in periportal regions (*Figure 2G*), and a dramatic reduction in proliferating KI67 positive hepatocytes (*Figure 2H*).

## Intrahepatic bile duct paucity in TAF4-null livers

In WT P5 liver, bile ducts were seen in both the hilum and peripheral regions (*Figure 3A*). In *Taf4a*[hep−/−] liver, bile ducts with a clear lumen were formed in the hilum, but their number rapidly decreases towards to periphery where residual cholangiocytes organised as bi-layered ductal structures that fail to undergo normal tubulogenesis (*Figure 3A*). Staining cholangiocytes at P12 with SOX9 showed normally formed mesenchyme-integrated bile ducts in the periportal regions of the hilum and periphery in WT liver (*Figure 3B*). Staining for CLDN3 marking the apical pole of the ductal cells revealed properly formed ducts with a clear lumen (*Figure 3—figure supplement 1B*). In mutant liver, SOX9 positive cholangiocytes were also observed, but even when located next to the large portal vein towards the hilum they did not form tubular ducts (*Figure 3B*) although CLDN3 staining revealed cell polarisation (*Figure 3—figure supplement 1B*). Staining with ACTA2 showed hepatic arteries adjacent to normal bile ducts in WT liver, which were absent in the peripheral regions of *Taf4a*[hep−/−] liver (*Figure 3B*). Portal vein mesenchyme was also significantly reduced in mutant liver (*Figures 3B and 1A*). Electron microscopy confirmed proper formation of bile ducts with lumen in WT liver, but disorganised ducts without lumen in *Taf4a*[hep−/−] liver (*Figure 3C*). The presence of ducts in the hilar region and their absence in the peripheral regions suggest that maturation is normally initiated during embryogenesis but is arrested upon TAF4 inactivation.

The above phenotype is similar to the loss of Notch signalling, associated with Alagille syndrome in humans, essential for normal bile duct numbers and morphogenesis (*Zong et al., 2009*). These similarities suggest that TAF4 may act as a cofactor for transcription factors of the Notch signalling pathway. In addition, the under-development of portal vein mesenchyme and hepatic arteries associated with defective bile duct morphogenesis supports the reported reciprocal nature of biliary, vascular, and mesenchyme maturation (*Hofmann et al., 2010*).

## Defective cell–cell contacts in TAF4-mutant liver

Tight junctions formed between neighbouring hepatocytes produce an impermeable blood–bile canaliculi barrier and establish the basolateral–apical axis. The apical membrane is covered with microvilli

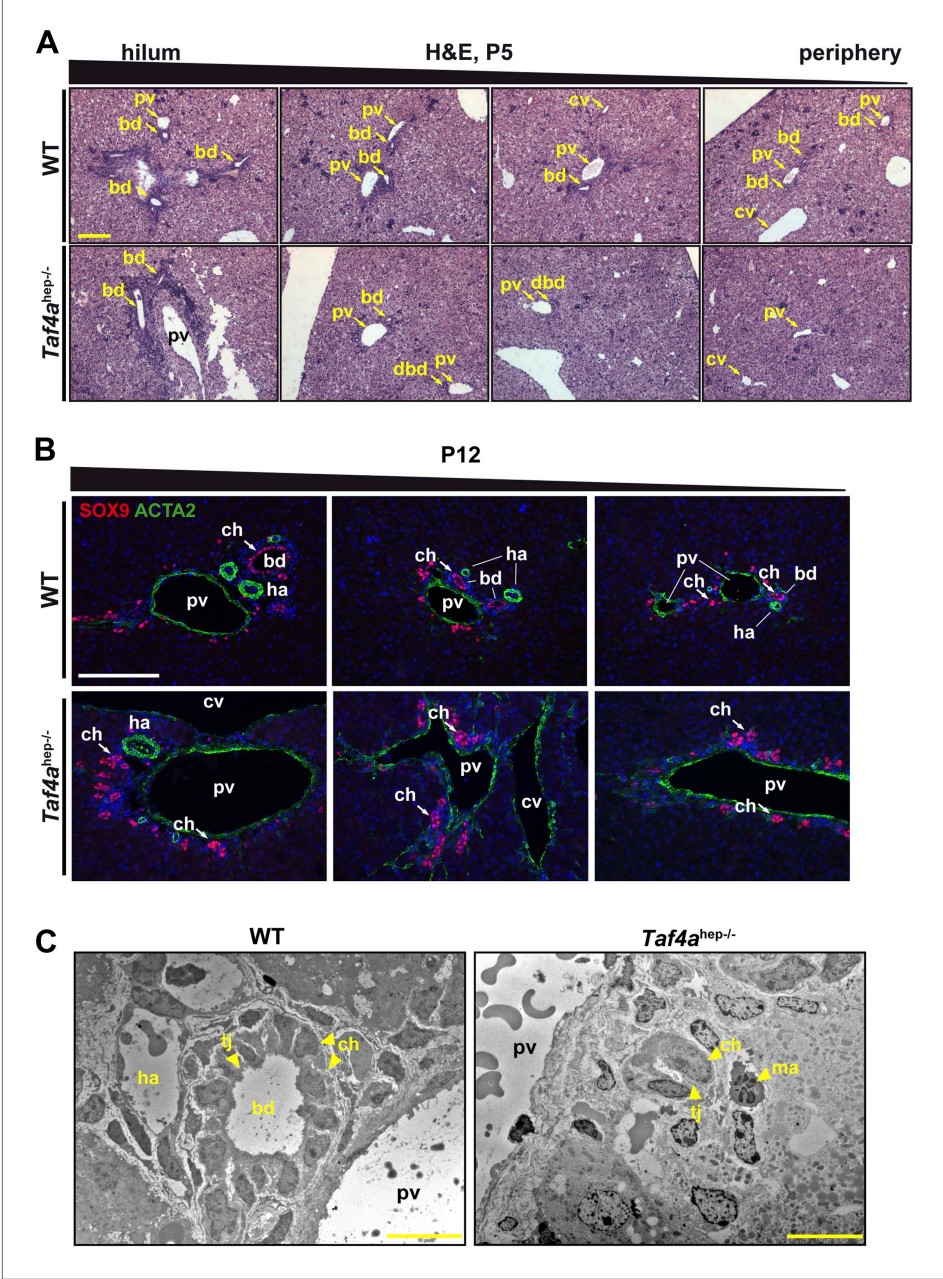

**Figure 3**. Bile duct paucity. (**A**) Hematoxylin–eosin staining of livers at P5 from the hilum towards the periphery as indicated (scale bar = 100 µm). (**B**) Immunofluorescence for SOX9 and ACTA2 in sections through WT and *Taf4a*[hep−/−] liver at P12 from the hilum towards the periphery as indicated (scale bar = 100 µm). (**C**) Electron micrographs illustrating bile ducts in WT and *Taf4a*[hep−/−] liver showing a normal duct with lumen in WT and defective ducts in the mutant lacking normal lumen (scale bar = 10 µm). pv; portal vein, bd; bile duct, cv; central vein, ha; hepatic artery, tj; tight junction, ch; cholangiocytes, ma; infiltrating macrophage.
The following figure supplement is available for figure 3:

**Figure supplement 1**. Defective tight junction formation and loss of bile–blood barrier.

in bile canaliculi, whereas the basolateral pole faces the blood flow in the sinusoid. *Taf4a*[hep−/−] liver displayed abnormal spaces between hepatocyte membranes indicative of impaired cell contacts and reduced number of tight junctions (***Figure 2G*** and ***Figure 3—figure supplement 1A***). CLDN3 and TJP1 staining showed extensive tight junctions all around the membrane of wild-type hepatocytes,

whereas, in the *Taf4a*<sup>hep–/–</sup> hepatocytes, staining was irregular and patchy, and many cells showed little or no staining (***Figure 3—figure supplement 1B–C***). Consequently, impaired cell–cell contacts and the absence of tubular bile ducts account for the observed accumulation of bile in the liver parenchyma and in the blood of the mutant animals.

## Gene expression changes in *Taf4a*<sup>hep–/–</sup> liver

We assessed changes in liver gene expression by RNA-seq at P12. In TAF4-mutant liver, 1408 genes were down-regulated and 776 up-regulated compared to wild-type (***Figure 4A***, and ***Figure 4A— figure supplement 1A*** and ***Supplementary file 1***, [***Alpern et al., 2014***]) The changes in expression of a selected set of genes were confirmed by RT-qPCR (***Figure 4B***). Down-regulated genes are predominately expressed in liver (endoderm), whereas the majority of up-regulated genes are annotated as enriched in brain, nervous system, and testis (***Figure 4A***).

Down-regulated genes showed significant enrichment for terms associated with liver metabolic functions (***Figure 4A—figure supplement 1B***) and provide insight to the observed phenotype. Genes of the neutral bile acid biosynthetic pathway (*Cyp39a1, Hsd3b7, Akr1d1, Cyp8b1, Slc27a5, Acox2, Scp2,* and *Baat*) were repressed as were the bile acid secretion and re-uptake components *Abcb1a* and *Abcg5/8*, respectively (***Figure 4A—figure supplement 1C***). Defective expression of these genes was reported to be associated with intrahepatic cholestasis and jaundice due to the production of abnormal cytotoxic bile or its impaired secretion (***Lefebvre et al., 2009***), all of which are seen in *Taf4a*<sup>hep–/–</sup> animals. *Cyp7a1* encoding an initiatory and rate-limiting enzyme responsible for up to 75% of total bile acid production was up-regulated. This may be explained by decreased expression of its transcriptional repressors SHP (*Nr0b2*) and FXR (*Nr1h4*), but also by repression of *Cyp8b1*. In *Cyp8b1*<sup>–/–</sup> mice, *Cyp7a1* expression is up-regulated due to loss of cholic acid, a primary bile acid produced by CYP8B1 (***Lefebvre et al., 2009***).

Many glycolysis/gluconeogenesis and fatty acid oxidation genes were repressed such as the glucose transporter GLUT2 (*Slc2a2*), *Gck*, the rate-limiting enzyme of glucose metabolism, *Pcx*, several glucose inter-conversion enzymes (*Pfkl, Pgk1, Pdha1, Eno1, Fbp1*), and *Aqp7/Aqp9*, important for glycerol supply and hepatic gluconeogenesis. Reduced expression of the components of all epithelial junction types: tight junctions (*Clnd1/2/3, Tjp3*), adherent junctions (*Cdh1, Pvrl1/2*), desmosomes (*Dsc2, Pkp1/3*), and gap junctions (*Gjb1/2*) supports the observed disruption of the hepatic epithelium and loss of the blood–bile barrier in the TAF4 mutant animals.

Several genes involved in glucose and lipid metabolism were up-regulated in *Taf4a*<sup>hep–/–</sup> livers, but surprisingly most encode enzymes that fulfil the complementary function in non-hepatic tissues. For example, *Ldhb, Eno2/3, Bpgm*, and *Fbp2* are normally expressed in heart, muscle, brain, or erythroid cells, but not in liver. Similarly, *G6pdx*, an enzyme of the pentose phosphate pathway, is normally highly expressed in adipose tissue but was up-regulated in *Taf4a*<sup>hep–/–</sup> livers. We also observed increased expression of fatty acid translocase Cd36 normally expressed in macrophages, muscle, and endothelium. Its up-regulation in liver is associated with insulin resistance, hyperinsulinaemia, and increased steatosis (***Miquilena-Colina et al., 2011***). The brain-specific isoforms of fatty acid-binding protein (*Fabp7*) and acyl-CoA-synthetase (*Acsl3*) that activates long chain fatty acids were also up-regulated. Activated FAs are shuttled across the mitochondrial membrane via carnitine palmitoyltransferase-1 that has three isoforms: liver, muscle, and brain. Expression of liver *Cpt1a* was unchanged, but the muscle *Cpt1b* isoform was overexpressed.

We compared our RNA-seq data with a multi-stage analysis of gene expression in developing liver (***Li et al., 2009***). From this data, we clustered more than 6600 genes with respect to their expression at E18.5, P7, and P14 comprising 805 of the 1408 down-regulated and 193 of the 776 up-regulated genes. Around 50% of down-regulated genes clustered with those showing low embryonic expression and induction by P7 (***Figure 4C***, clusters 1 and 2). In contrast and despite the fact that only a minority of up-regulated transcripts were present in the Li et al. data, they are most represented amongst genes whose expression is high at E18.5 and down-regulated between P7 and P14 (***Figure 4C***). This conclusion was confirmed when considering the mean neonatal expression value for up- and down-regulated genes relative to their expression at E18.5. The mean expression of down-regulated genes increased after birth in WT, whereas that of the up-regulated genes decreased (***Figure 4D***).

Interestingly, several imprinted genes (*Dlk1, Meg3, Dio3, Plagl1, Peg3, Peg10, Igf2as, Airn, Igf2r, Malat1, Neat1,* and *Grb10*) were up-regulated in mutant liver. These genes regulate fetal growth and

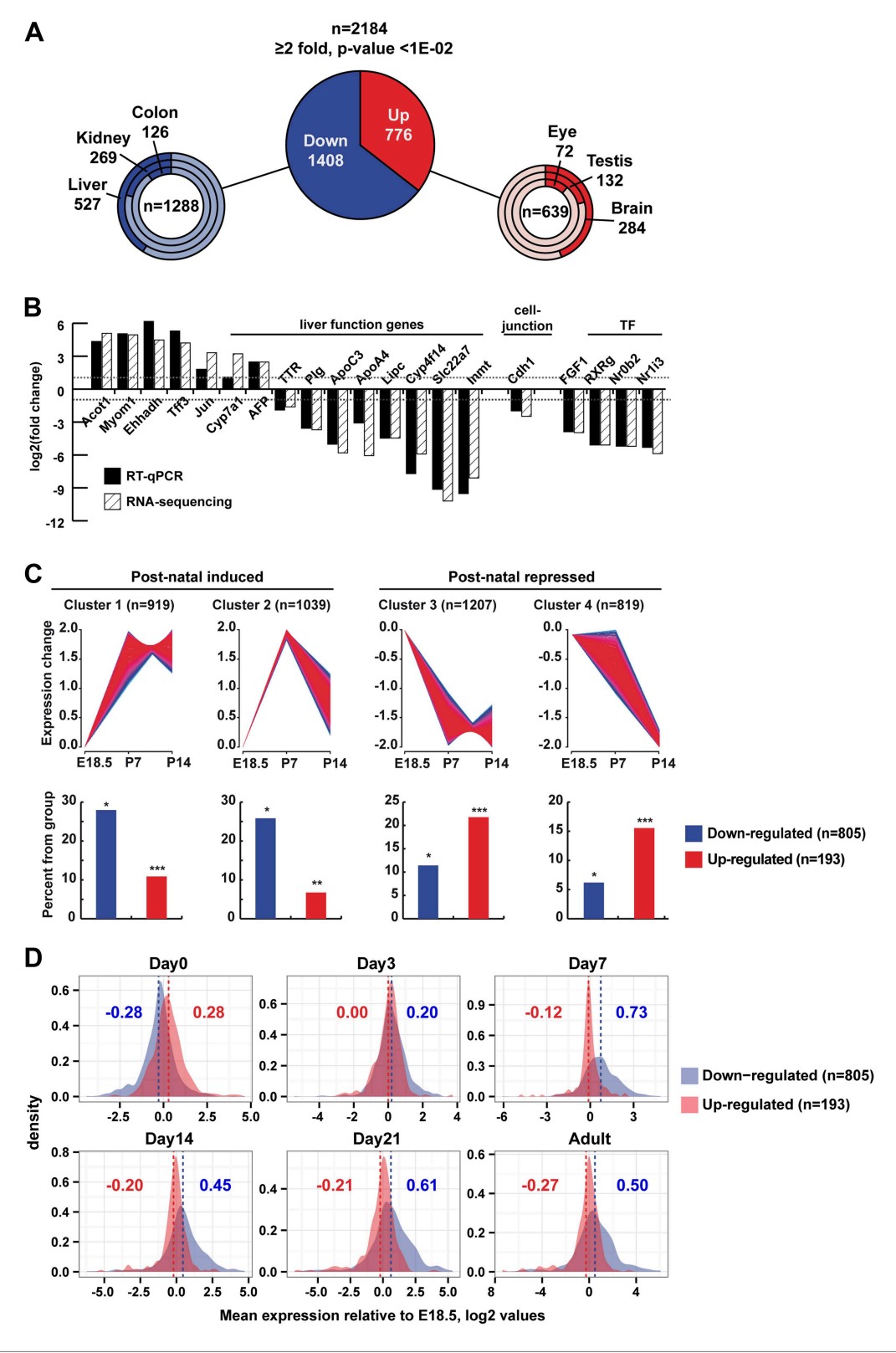

**Figure 4**. Gene expression. (**A**) Ontology of up- and down-regulated genes. (**B**) Comparison of gene expression changes measured by RNA-seq and qPCR. (**C**) Assignment of deregulated transcripts to clusters determined by their expression kinetics during normal liver development at the indicated stages. Gene expression data for last
*Figure 4. Continued on next page*

*Figure 4. Continued*

day of embryonic development (E18.5), P7 and P14, was normalized to have a mean of 0 and a SD of 1 and clustered using fuzzy c-means clustering (*Futschik and Carlisle, 2005*) implemented in the *mfuzz* R package. The number of clusters was decided empirically. Each cluster was then overlapped with the down- and up-regulated genes from our RNA-seq. The significance of cluster assignments was assessed by calculation of the hypergeometric p-value. The number of up- and down-regulated genes included in the public data is indicated along with the % of each class present in the cluster (hypergeometric tests p-values: *p < 1e−06, **p < 1e−04, ***p < 0.05). (**D**) The Log2 mean expression value during neonatal stages for up- and down-regulated genes relative to their expression at E18.5. The mean expression of down-regulated genes increased after birth, whereas that of the up-regulated genes decreased.

The following figure supplement is available for figure 4:

**Figure supplement 1**. Deregulated gene expression in *Taf4a*[hep−/−] liver.

are post-natal repressed. Expression of these genes is a hallmark of immature tissue, showing that TAF4 is required for normal post-natal hepatoblast maturation.

## Decreased expression of TFIID components and impaired TFIID formation in *Taf4a*[hep−/−] liver

We investigated expression of TFIID components in *Taf4a*[hep−/−] liver. TAF4 expression was strongly decreased in extracts from P12 *Taf4a*[hep−/−] liver (*Figure 5A*) but not totally lost since its expression persisted in the non-hepatocyte cells of the liver. Expression of TBP and several TAFs was also diminished while little change was observed for TFIIB, TFIIE, Pol II, and HNF4A (*Figure 5B*). We performed anti-TBP immunoprecipitation (IP) and normalised the amount of IP TBP from WT and *Taf4a*[hep−/−] liver. Under these conditions, diminished amounts of all tested TAFs were observed in the TBP IP from *Taf4a*[hep−/−] (*Figure 5C*). Thus, consistent with the critical role of TAF4 in the TFIID core complex (*Wright et al., 2006*, *Bieniossek et al., 2013*), loss of TAF4 led to reduced TBP and TAF accumulation likely due to the disassembly of TFIID witnessed by the reduced co-precipitation of TAFs with TBP. No significant expression of TAF4b that can replace TAF4 in the core complex was seen in extracts from WT and *Taf4a*[hep−/−] liver and in the respective TBP IPs (*Figure 5D* and data not shown). TAF4b therefore cannot substitute for TAF4 to maintain TFIID integrity in this tissue.

## Decreased pre-initiation complex formation at genes down-regulated in *Taf4a*[hep−/−] liver

We performed ChIP-seq from wild-type and *Taf4a*[hep−/−] P12 liver using antibodies against TBP, TAF3, TFIIB, TFIIE, RNA polymerase II (Pol II), trimethylated lysine 4 of histone H3 (H3K4me3), and CTCF to monitor PIC formation and chromatin organisation.

The TBP, TFIIB, and TFIIE ChIP-seq data indicated a global decrease in their genomic occupancy and a 2–2.5 fold lower occupancy at the TSS of expressed genes in *Taf4a*[hep−/−] liver (*Figure 6A*). No comparable decrease was seen for CTCF whose distribution is largely unchanged (*Figure 6B*). A much stronger decrease in PIC formation was seen at the TSS of down-regulated genes. Similarly, an almost two-fold reduction of promoter-proximal paused and elongating Pol II was seen at all expressed genes, with a much stronger reduction at down-regulated genes (*Figure 7A*). For example, PIC formation, Pol II recruitment, and H3K4me3 were lost at the TSS of *Dio1* that is post-natal induced in WT (*Figure 6C*). Similar results were seen at many other post-natal activated liver function genes indicating that loss of their expression in *Taf4a*[hep−/−] liver corresponds to defective PIC formation in the absence of TAF4.

## Decreased Pol II pausing at genes up-regulated in *Taf4a*[hep−/−] liver

More intriguingly, diminished PIC formation was also observed at up-regulated genes with reduced paused Pol II downstream of the TSS (*Figure 7A*). The Pol II meta-profile shows, however, that levels of elongating Pol II were comparable in WT and mutant indicating reduced Pol II pausing and a relative increase in elongating Pol II in *Taf4a*[hep−/−] liver. This is exemplified at the *Angptl4* gene where equivalent amounts of PIC and paused Pol II were seen in WT and mutant, whereas a higher density of elongating Pol II was observed in *Taf4a*[hep−/−] liver (*Figure 6C*). Similar results were observed at the *Malat1*, *Txnip*, *Tat,* and *Cyp7a1* genes (*Figure 7B–C*). At *Ehhadh*, an increase in both paused and elongating Pol II was observed. Up-regulated *Afp* expression also reflects an overall increase in Pol II occupancy. ChIP-qPCR

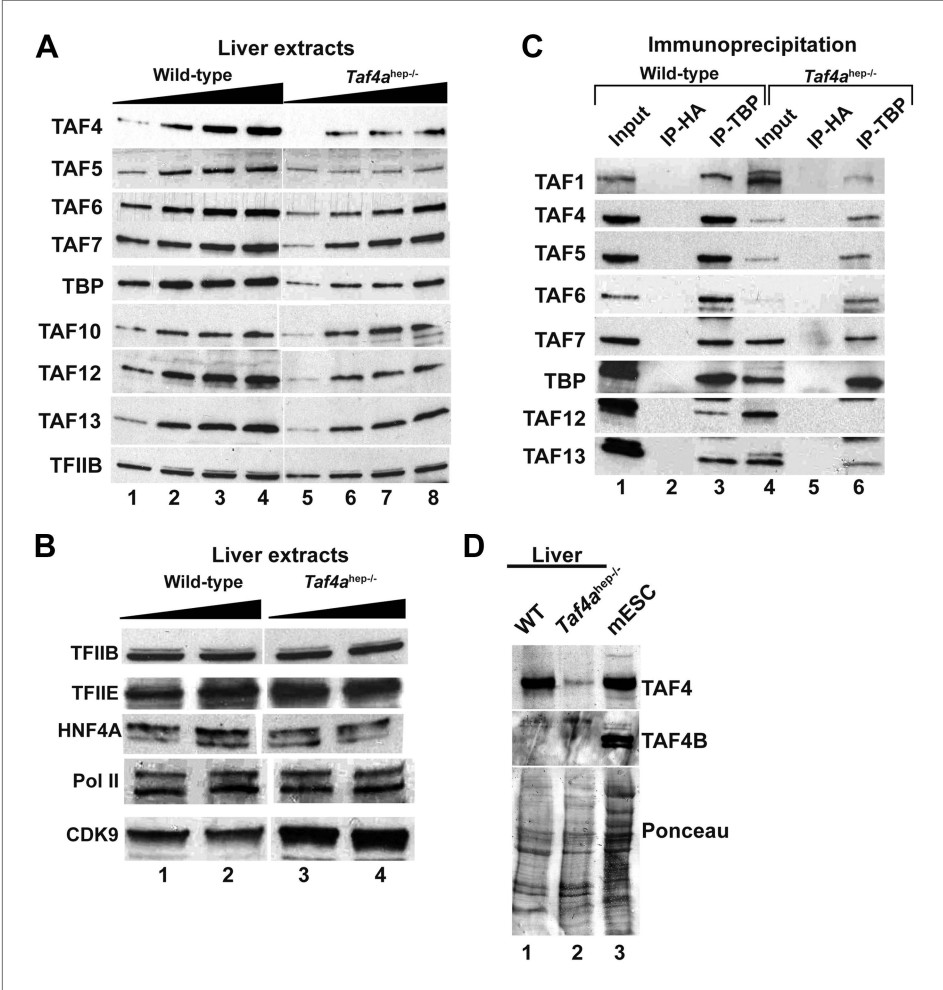

**Figure 5**. Expression and integrity of TFIID. (**A** and **B**) Expression of TBP and TAFs in liver nuclear extracts (10, 20, 30, 40 µg in **A** and 20 and 40 µg in **B**). (**C**) Expression of the indicated TAFs in the input, the control anti-HA IP, and the anti-TBP IP. (**D**) Absence of TAF4B in the wild-type and TAF4-mutant liver extracts. An extract from mouse embryonic stem cells (mESC) was used as a positive control where TAF4B can be clearly detected.

indicated enhanced recruitment of the CDK9 subunit of the positive regulator of elongation PTEFb at the TSS, gene body, and 3'UTR of the *Angptl4, Malat*, and *Ehhadh* genes in the mutant, whereas no such increase was seen at down-regulated genes that displayed a generally low level of CDK9 (*Figure 7D*).

These results reveal two distinct mechanisms for gene regulation in *Taf4a*[hep−/−] liver. Down-regulation is due to defective PIC formation, while up-regulation of many genes reflects decreased pausing and increased elongating Pol II and CDK9 recruitment.

## TBP-independent TAF3 genome occupancy

TAF3 was recruited to the TSS as expected for a TFIID component, but TAF3 occupancy and the levels of H3K4me3 at the TSS were much less affected than TBP occupancy and PIC formation (*Figure 6A*). TAF3 occupancy rather correlated not with TBP and PIC formation but with H3K4me3 in agreement with the idea that it can be recruited independently of TBP and TFIID via interaction of its PHD domain with this mark (*Vermeulen et al., 2007*; *Lauberth et al., 2013*). Moreover, TAF3 does not precisely co-localise with TBP at the TSS. TBP and TFIIB localised immediately upstream of TFIIE, whereas TAF3 localised downstream with paused Pol II (*Figure 6—figure supplement 1A*). Reanalysis of public ES cell data (*Liu et al., 2011*) also showed localisation of TAF3 and TAF1 downstream of the TSS, while TBP localised upstream. This differential crosslinking provides insight into how TFIID interacts with promoter DNA in the PIC.

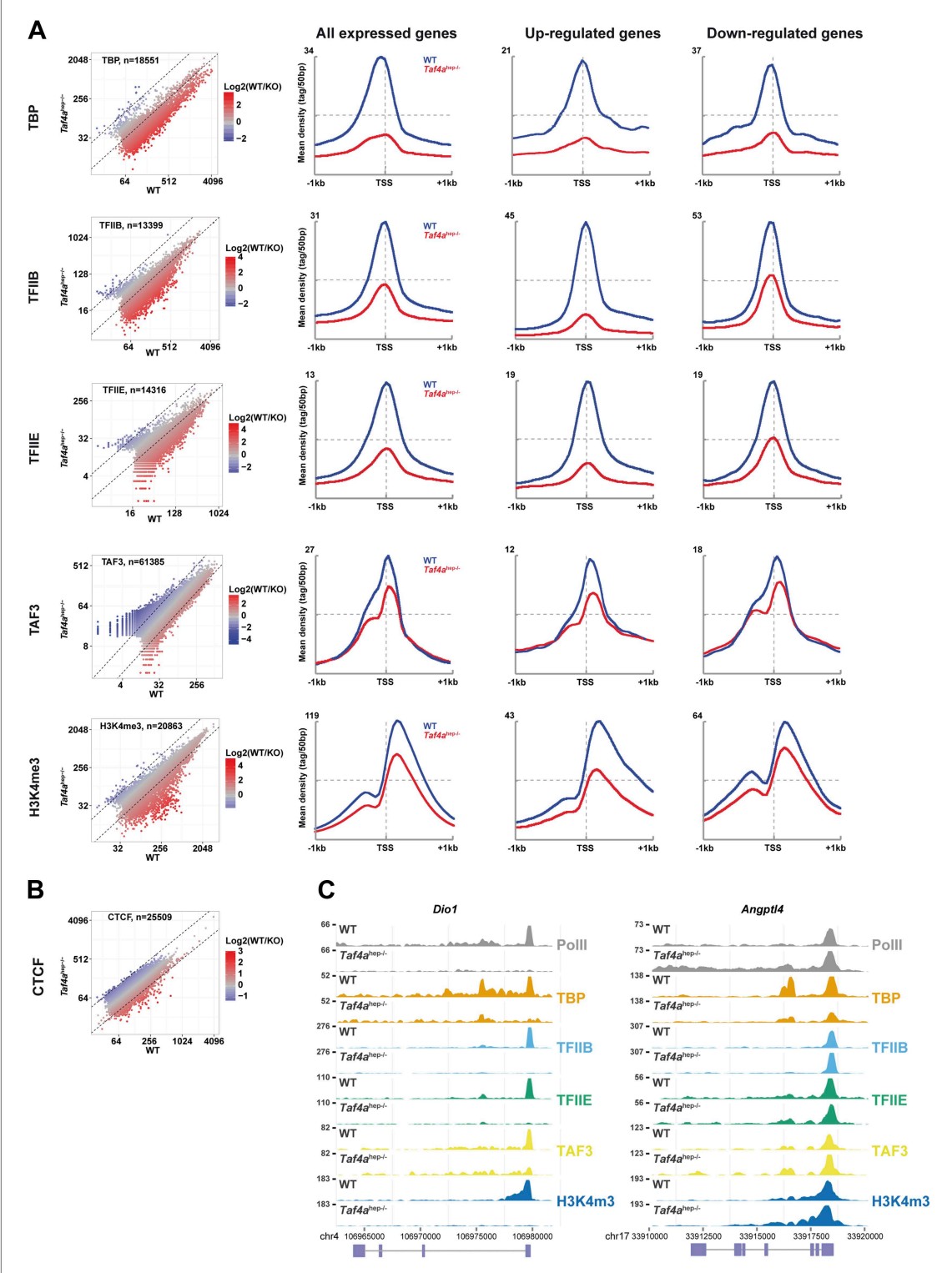

**Figure 6**. Defective pre-initiation complex formation. (**A** and **B**) Genomic occupancy of the indicated factors and of H3K4me3 in WT and *Taf4a*hep−/− liver. Left panel shows the global profile and the right panels show occupancy at all expressed genes or up- and down-regulated genes as indicated. (**C**) Integrated read count of the indicated ChIP-seq tracks and the *Dio1* or *Angptl* loci.

The following figure supplement is available for figure 6:

**Figure supplement 1**. TAF3 genomic occupancy in liver and ES cells.

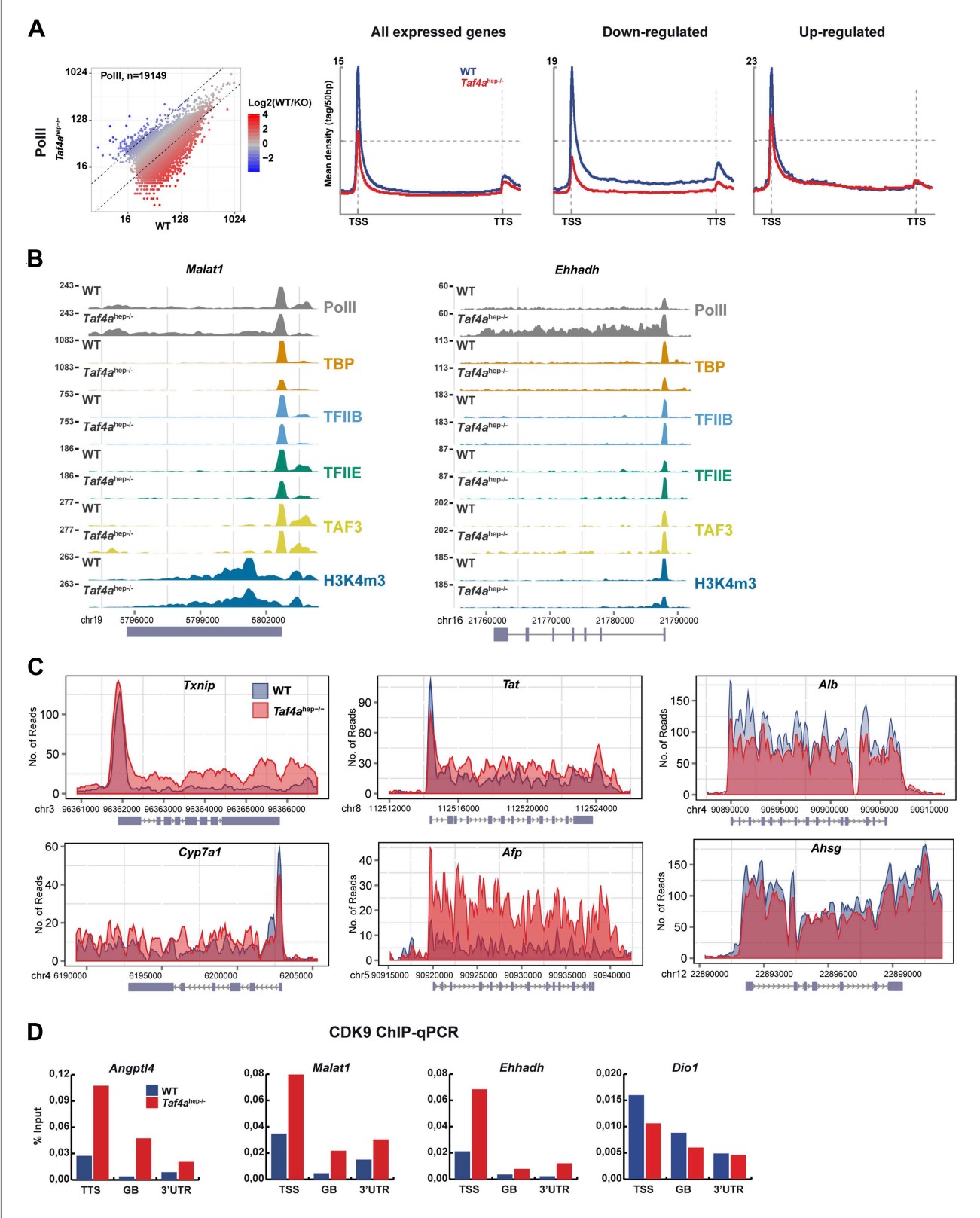

**Figure 7**. Changes in Pol II elongation. (**A**) Pol II genomic occupancy in WT and in *Taf4a*^hep−/− liver. (**B**) ChIP-seq tracks at the *Malat* or *Ehhadh* loci. (**C**) Overlays of Pol II ChIP-seq densities at the indicated gene loci. (**D**) ChIP-qPCR of CDK9 at the TSS, gene body (GB), and 3'UTR of the indicated genes. The % input is indicated.

TAF3 occupied a large set of distal loci in the absence of TBP and PIC (*Figure 6—figure supplement 1B*), as previously reported in ES cells (*Liu et al., 2011*). In ES cells, 30% of distal TAF3 sites correlated with CTCF occupancy, whereas in liver only 11% correlation was seen (*Figures 6—figure supplement 1B,C*). This is not due to differences in analysis procedures, as we independently calculated a 29–31% TAF3–CTCF correlation from the public ES data set (*Figure 6—figure supplement 1C*). In hepatocytes, the TAF3–CTCF correlation was higher at the TSS (0.55) than at distal regions, the opposite of ES cells. A high TAF3–H3K4me3 correlation was observed at distal sites suggesting that this may be a mechanism of recruitment to these sites. Nevertheless, H3K4me3 levels are low at many of these distal TAF3 sites and other recruitment mechanisms cannot be excluded. Comparison of TAF3 genomic occupancy in liver and ES cells identified common and specifically occupied sites at the TSS, reflecting different sets of active promoters in these two cell types. Moreover, many of the distal regions are also unique to each cell type (*Figure 6—figure supplement 1D*). Together these data indicate that TAF3 can be recruited to the genome independently of TBP and PIC formation not only at distal regions but also at the TSS.

We also noted that TAF3 and H3K4me3 levels did not fully correlate with Pol II occupancy and PIC formation at the most highly expressed genes. TSS occupancy by TBP, TFIIB, TFIIE, and Pol II is positively correlated with gene expression where highest expressed genes showed highest occupancy (*Figure 6—figure supplement 1E*). In contrast, H3K4me3 and TAF3 displayed reduced occupancy at this small highly expressed genes subclass, exemplified by *Afp*, *Alb,* and *Ashg*, with little Pol II pausing and high levels of elongating Pol II (*Figure 7C*). Thus TAF3 and H3K4me3 show reduced occupancy at highly expressed genes lacking paused Pol II.

## TAF4 promotes HNF4A occupancy of conserved and functional regulatory elements

Nuclear receptor HNF4A is a major regulator of hepatocyte gene expression (*Hayhurst et al., 2001*; *Parviz et al., 2003*). In WT and *Taf4a*[hep−/−] liver, HNF4A occupied >68,000 binding sites enriched around the TSS and comprising a motif essentially identical to that previously defined (*Figure 8A*). Nevertheless, TAF4 loss modified HNF4A genomic occupancy as >7100 sites were depleted in the mutant, exemplified by the *Dio1* and *Slc22a7* loci (*Figure 8—figure supplement 1A*), and >4100 sites were enriched in the mutant. Globally, sites depleted in the mutant localised close to the TSS (*Figure 8B*) and those associated with 487 down-regulated genes, using a window of ±40 kb with respect to the TSS, define a subpopulation closest to the TSS. In contrast, no TSS enrichment was seen for the small number of depleted sites associated with up-regulated genes (red in *Figure 8B*). Unlike the sites depleted in the mutant, enriched sites associated with regulated genes showed no strong localisation at the TSS (*Figure 8C*). A small number of these sites were however associated with up-regulated genes, perhaps corresponding to sites occupied during embryogenesis and lost in wild-type neonatal liver but not in the mutant.

Not all 68,000 detected HNF4A binding sites are functional in terms of gene regulation. At the *Dio1* gene for example, two HNF4A binding sites were lost in the mutant, but a third was unchanged (*Figure 8—figure supplement 1A*). We therefore asked if the HNF4A sites depleted in the mutant correspond to functionally active sites. Comparison with public adult liver data showed that many depleted HNF4A sites associate with p300, H3K4me1, and H3K27ac, whereas less of the enriched HNF4A sites show this association (*Figure 8—figure supplement 1B*). Previous experiments identified evolutionary conserved HNF4A sites, suggesting they are critical for function (*Schmidt et al., 2010*). A subset of HNF4A occupied sites is associated with binding sites for CEBPA, HNF6, and FOXA2 forming cis-regulatory modules (CRMs) many of which are conserved in at least two species (*Ballester et al., 2014*). Other sites, designated here as 'singletons', correspond to HNF4A occupied sites that are not associated with binding sites for these other factors, although some are also conserved between at least two species. 60% of HNF4A sites depleted upon TAF4 inactivation overlap with CRMs, 16% of which are conserved (*Figure 8D*). These CRMs display a higher proportion of sites for three or four factors and many associate with p300, H3K4me1, and H3K27ac (*Figure 8—figure supplement 1B*). In contrast, only 20% of HNF4A sites enriched in the mutant overlap with CRMs of which only 4% are conserved and they show lesser association with active marks. Furthermore, CRMs depleted in the mutant, and in particular those associated with the 487 down-regulated genes, are strongly enriched at the TSS, whereas those with enhanced HNF4A occupancy appear more random although they are so few that there is no statistical significance (*Figure 8E*). This allowed us to define

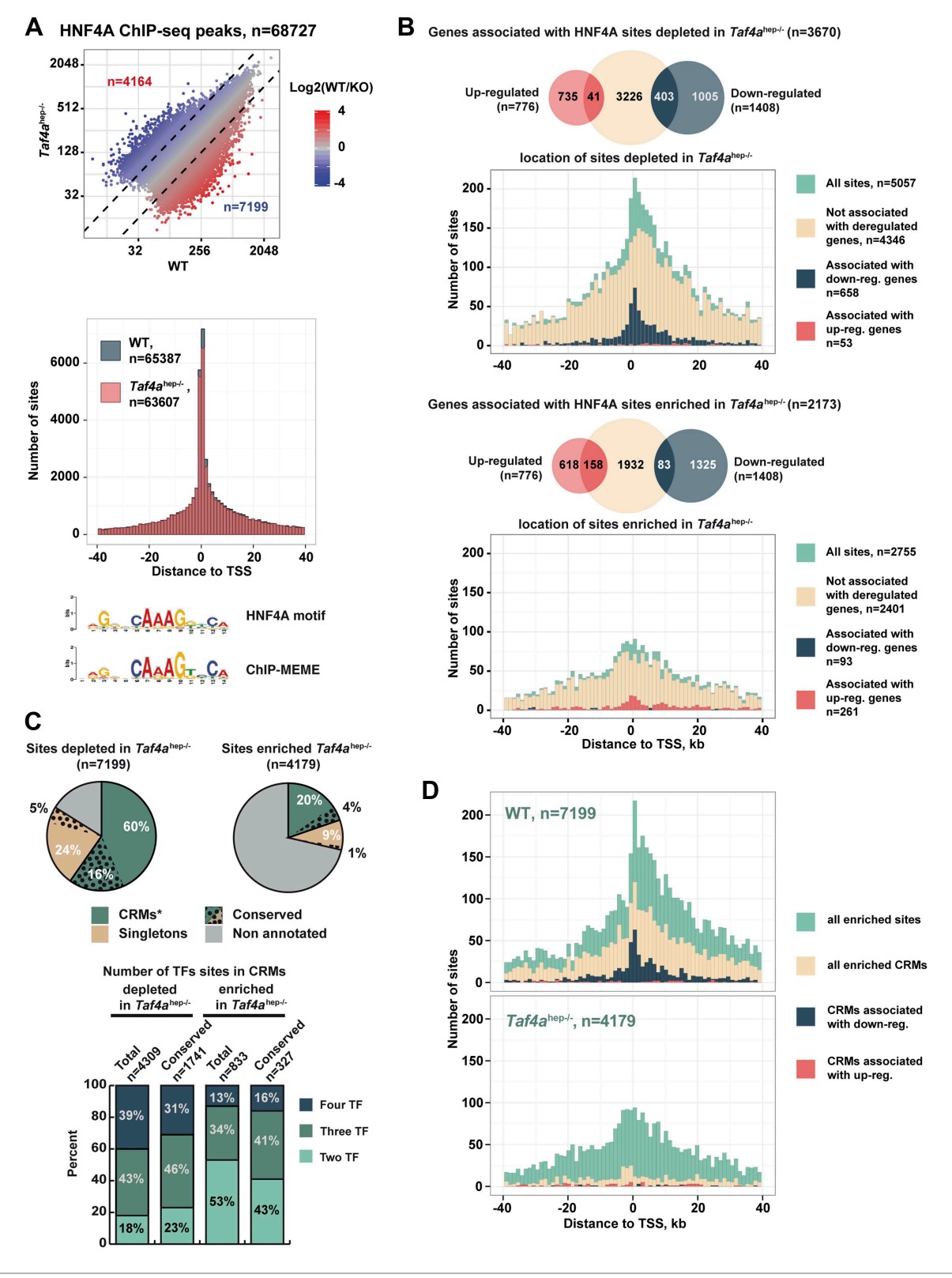

**Figure 8**. TAF4 is required to recruit HNF4A to functional CRMs. (**A**) Comparison of HNF4A ChIP-seq in WT and in *Taf4a*hep−/− liver and location of HNF4A occupied sites relative to the TSS. Comparison of the HNF4A consensus-binding sequence from our data generated by ChIP-MEME with the previously defined sequence. (**B**) Venn diagrams illustrate the number of genes with at least one HNF4A binding site either depleted or

*Figure 8. Continued on next page*

*Figure 8. Continued*

enriched in *Taf4a*[hep−/−] liver, within ±40 kb with respect to the TSS, intersected with up- or down-regulated genes. Graphs illustrate the locations of either the total, depleted or enriched HNF4A sites associated with the up and down-regulated genes within a window of ±40 kb with respect to the TSS. Total sites are shown in green and the sites associated with up- and down-regulated genes are shown in red and blue, respectively. (**C**) Upper panel shows the % of HNF4A-occupied sites enriched or depleted in *Taf4a*[hep−/−] liver that correspond to evolutionarily conserved or non-conserved CRMs. The lower panel shows the % of CRMs in each class that comprise sites for 1, 2, or 3 transcription factors in addition to HNF4A. (**D**) Location of HNF4A-occupied sites or CRMs, as indicated, enriched in WT or *Taf4a*[hep−/−] livers relative to the TSS corresponding to all peaks, all CRMs, or CRMs associated with down- and up-regulated genes.

The following figure supplements are available for figure 8:

**Figure supplement 1**. HNF4A genomic occupancy.

**Figure supplement 2**. HNF4A genomic occupancy during liver development.

a set of 296 down-regulated genes where the depleted HNF4A sites are localised between ±10 kb relative to the TSS (*Supplementary file 1*). ChIP-qPCR showed occupancy of CRMs at the *Dio1*, *Slc22a7*, and *Car5a* genes by FOXA2, HNF6, and CEBPA in WT liver, but not in the TAF4 mutant (*Figure 8—figure supplement 1C*). Conversely, at one of the rarer CRMs enriched in the mutant, ChIP-qPCR showed increased HNF4A, CEBPA, and HNF6 occupancy, while that of FOXA2 was not significantly modified. TAF4 is therefore required for occupancy of a set of functional CRMs located close to the TSS explaining the failure to activate HNF4A-regulated liver function genes in the TAF4 mutant.

We examined HNF4A genomic occupancy at E18.5 compared to neonatal and adult liver profiles. 4353 sites were occupied at E18.5 compared to >65,000 at P12 and >70,000 at P75 (*Figure 8—figure supplement 2A*). The reduced HNF4A occupancy at E18.5 is not due to the lack of its expression that is comparable at E18.5 and adult stages (*Figure 8—figure supplement 2B*). The E18.5-occupied sites associate with >1800 genes enriched in liver metabolic functions (*Supplementary file 1*). While almost all E18.5-occupied sites were occupied at P12 and P75, subsets of sites show stronger occupation at P12 or P75 compared to other stages. We investigated the occupancy of sites enriched or depleted in the TAF4 mutant at the different developmental stages. A significant proportion of depleted sites maps to those with enriched occupancy at the neonatal stage (*Figure 8—figure supplement 2C*). In contrast, around 30% of sites enriched in the mutant show only very low occupancy at other stages. Thus in the absence of TAF4, HNF4A occupies a set of sites the vast majority of which are not normally occupied either in embryonic or adult liver. This suggests that there may be a disorganisation of the chromatin landscape in the TAF4-mutant liver that allows HNF4A accessibility to sites that are not accessible in wild type. TAF4 is therefore necessary to promote normal HNF4A genomic occupancy, in particular of promoter-associated functional CRMs.

## HNF4A interacts with the TAF4–TAF12 heterodimer

It has previously been shown that HNF4A interacts physically with TFIID via TBP (*Takahashi et al., 2009*). HNF4A correlated with TBP/PIC occupancy at the TSS where HNF4A sites are enriched, but also at distal sites, indicative of functional HNF4A–TFIID interactions in vivo in hepatocytes (*Figure 6—figure supplement 1C*). Diminished TBP recruitment in TAF4 mutant liver may by itself not explain the loss of PIC formation and HNF4A occupancy of the functional TSS-proximal CRMs. We therefore asked whether HNF4A also interacts with TAF4, as loss of this interaction would better explain the observed phenotype. HEK cells were transfected with vectors expressing combinations of HNF4A, full-length mouse, or human TAF4 in the presence or absence of TAF12 and the extracts precipitated with anti-HNF4A antibodies. TAF4 alone and the TAF4–TAF12 heterodimer efficiently co-precipitated with HNF4A compared to the negative controls (*Figure 9A*, lanes 2, 3, and 5 compared to 1 and 4). We also expressed HNF4A along with the C-terminal histone fold-containing region of TAF4. The TAF4(805–1083)–TAF12 heterodimer efficiently co-precipitated with HNF4A compared to the negative control (lanes 8 and 6), however, TAF4(805–1083) alone did not co-precipitate with HNF4A (lane 7) as this short domain is mainly insoluble in the absence of TAF12. As a further control, no co-precipitation of

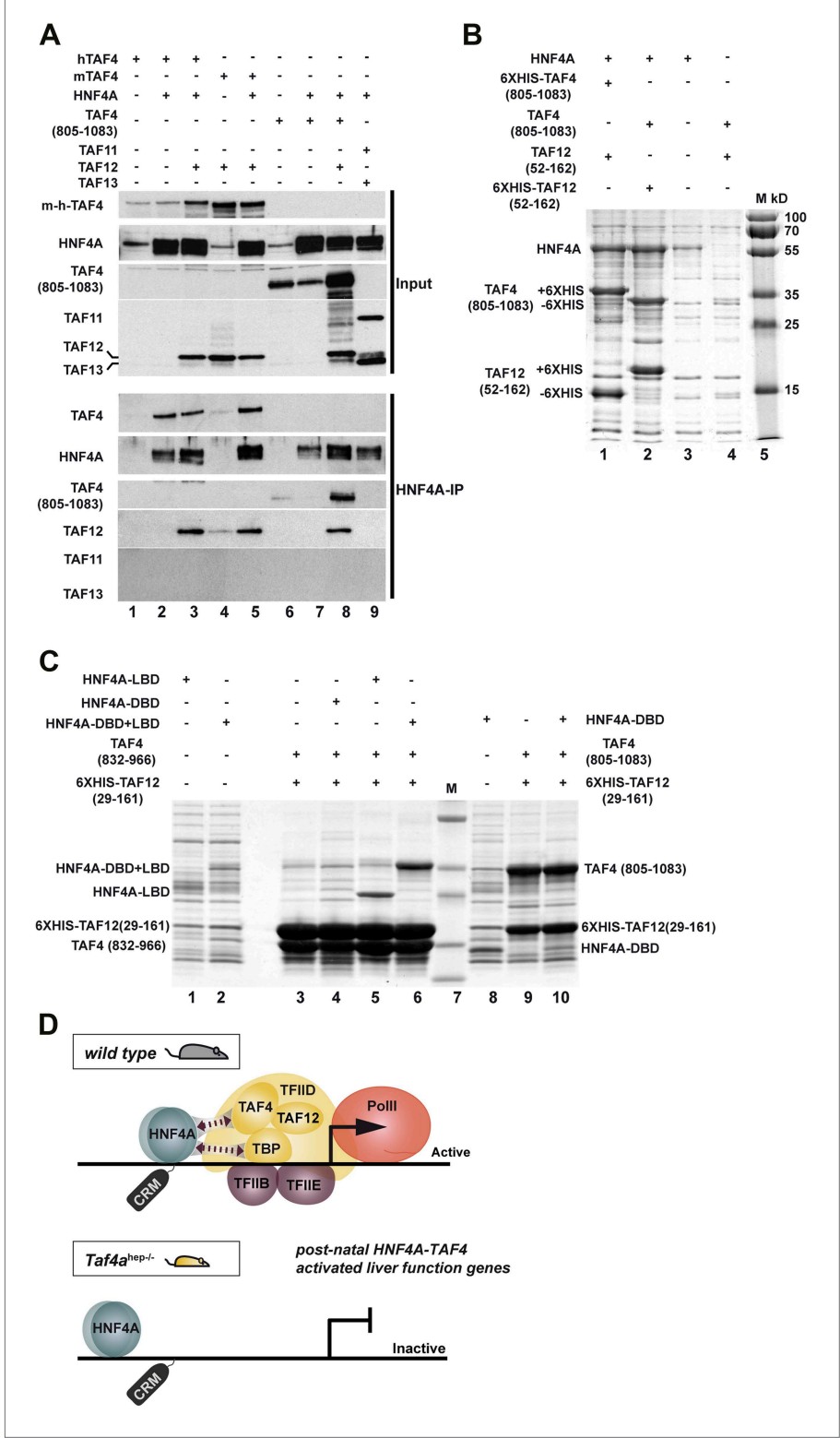

**Figure 9**. HNF4A interacts directly with the TAF4–TAF12 heterodimer via its LBD. (**A**) HEK cells were transfected with vectors expressing the constructs indicated above each lane. The upper panel represents proteins in the transfected cell extracts, the lower the proteins in the anti-HNF4A IP. (**B**) Formation of an HNF4A–TAF4–TAF12 complex from bacterial expressed proteins. SDS-PAGE followed by Coomassie brilliant blue staining of proteins

*Figure 9. Continued on next page*

*Figure 9. Continued*

retained on the cobalt-agarose column. Co-expressed proteins are shown above each lane and their locations to the left of the panel. Similar experiments with N-terminal 6HIS-tagged HNF4A were also attempted, but the presence of the tag induced degradation of recombinant HNF4A in bacteria not shown. (**C**) Bacterial co-expression of HNF4A domains with TAF4–TAF12. Co-expressed proteins are shown above each lane and their locations to the left of the panel. As the HNF4A-DBD co-migrates with TAF4 (832–966), it was re-expressed with TAF4 (805–1083). (**D**) Model for cooperative HNF4A–TFIID function in hepatocyte gene activation. HNF4A interacts with TBP and TAF4 in TFIID via its DBD and LBD, respectively. These interactions are required for PIC formation on target promoters and occupancy of functional HNF4A sites regulating transcription of liver-specific genes during neonatal hepatic maturation.

HNF4A with the TAF11–TAF13 histone fold pair was observed (lane 9). HNF4A therefore specifically interacts with the TAF4–TAF12 heterodimer.

To confirm this interaction, we co-expressed TAF4(amino acids 805–1083)–TAF12(29–161), containing their histone fold regions that form a soluble heterodimer in *Escherichia coli* (*Thuault et al., 2002*), with full length HNF4A. HNF4A was expressed with the 6HIS-tagged TAF4–TAF12 heterodimer and the proteins were purified over a cobalt agarose column and visualised by SDS-PAGE and Coomassie blue staining. HNF4A co-purified in close to stoichiometric amounts with the TAF4–TAF12 heterodimer (*Figure 9B*), while only trace amounts were non-specifically retained on the column in the absence of tagged TAF4–TAF12. To better characterise this interaction, we co-expressed TAF4–TAF12 with either the DNA binding domain of HNF4A (DBD, 55–135), the DBD-hinge-ligand binding domain (DBD–LBD, 55–377), or the LBD alone (148–377). These constructs were co-expressed with 6XHIS-tagged TAF12 (25–160) and a deletion of TAF4 (83–966) comprising the HFD. Both the DBD–LBD and the LBD alone were strongly retained on the column only in the presence of TAF4–TAF12 (*Figure 9C*, lanes 5 and 6 compared to 1 and 2). As the DBD alone co-migrates with TAF4 (832–966) and is not visible, we expressed the DBD alone with the longer version of TAF4 (805–1083). The DBD was not specifically retained in the presence of TAF4–TAF12 (lanes 8–10). These data indicate that HNF4A directly forms a complex with the TAF4–TAF12 heterodimer via its LBD.

## Discussion

### TAF4 and HNF4A cooperatively regulate hepatocyte gene expression

TAF4 inactivation in early post-natal hepatocytes led to defective liver organisation and function resulting in death of the animals by P15. Three major features characterise this phenotype: defective bile duct formation, disorganised hepatocyte epithelium with loss of cell junctions and blood–bile barrier, lack of post-natal gene activation and consequent metabolic abnormalities. Many of these features, indicative of retarded hepatocyte maturation, can be explained by TAF4 acting as a cofactor for HNF4A, an idea that is supported by functional and biochemical data.

Several features of the TAF4 knockout are observed when HNF4A is inactivated during hepatoblast development, such as disorganisation of the hepatocyte epithelium and liver architecture, loss of cell junctions, and glycogen accumulation (*Parviz et al., 2003*). While the genes involved in these processes are activated by HNF4A during embryogenesis before TAF4 inactivation, TAF4 is required to maintain their expression as PIC formation and Pol II recruitment at their TSS were diminished at post-natal stages in mutant liver. At most of these genes, however, HNF4A occupancy was not strongly affected by TAF4 inactivation.

HNF4A is required for post-natal expression of many liver metabolic function genes (*Hayhurst et al., 2001*; *Inoue et al., 2006*; *Kyrmizi et al., 2006*). For example, HNF4A is necessary for fatty acid metabolism (*Martinez-Jimenez et al., 2010*) and bile acid synthesis (*Inoue et al., 2006*). In these studies, however, HNF4A inactivation occurred between P35 and P40, while using our mice (carrying a different *Alb*-Cre transgene) TAF4 inactivation is complete by P12 resulting in the loss of HNF4A binding to regulatory elements of at least 296 genes. Consequently, we observe a more severe general loss of liver metabolic functions with many relevant genes amongst these 296 requiring TAF4–HNF4A cooperation. While loss of HNF4A in adults has little effect on the expression of other nuclear receptors such as FXR, PXR, and LXR that play important functions in liver, their expression is strongly diminished in the *Taf4a*[hep−/−] liver also contributing to the more severe phenotype.

Two different situations can thus be defined. At many, but not all, genes activated by HNF4A pre-natally in the presence of TAF4, subsequent TAF4 inactivation leads to diminished PIC formation, but has little effect on HNF4A occupancy of their regulatory elements that persists at post-natal stages. In contrast, TAF4 is required to stably recruit HNF4A to regulatory elements and for PIC formation at post-natal activated genes as both processes are disrupted upon TAF4 inactivation.

Our data also define for the first time the full scope of HNF4A in maintaining the integrity of the hepatocyte epithelium acquired during embryogenesis and in activation of the post-natal gene expression programme. In previous studies not all of these functions were observed due to the inappropriate timing of HNF4A inactivation that was either too early or too late to reveal them. While loss of HNF4A function accounts for many aspects of the TAF4 phenotype, we do not exclude that TAF4 may act as a cofactor for other transcription factors, for example RPBJ-NCID involved in bile duct morphogenesis.

In addition to the above functional data, biochemical data also support the idea that TAF4 acts as a cofactor for HNF4A. Epitope-tagged HNF4A can precipitate TFIID (*Takahashi et al., 2009*). We also show a correlation between HNF4A, TBP/TFIID, and PIC occupancy at distal sites, possibly enhancers, indicative of functional HNF4A–TFIID interactions in vivo. While this may in part be accounted for by interaction of the HNF4A-DBD with TBP (*Takahashi et al., 2009*), we show that HNF4A also interacts with the TAF4–TAF12 heterodimer via its LBD. HNF4A directly forms a complex with the histone fold-containing regions of TAF4–TAF12 when co-expressed in bacteria. In the absence of TAF4, HNF4A occupancy of TSS-proximal CRMs and/or PIC formation are compromised strongly suggesting that two direct HNF4A–TFIID interactions via the TAF4–TAF12 heterodimer and TBP are required to activate liver function genes (*Figure 9D*). An analogous mechanism has been described where TAF4 acts as a cofactor for E-proteins (*Chen et al., 2013*). E-proteins interact with the TAFH domain of TAF4 as opposed to the histone-fold region for HNF4A, and this interaction enhances TFIID binding to the core promoter. Our data are consistent with a model where stable binding of HNF4A to TSS proximal CRMs and PIC formation require HNF4A–TFIID interactions. The concomitant loss of PIC formation and CRM occupancy upon TAF4 inactivation indicates that they are mutually dependent events.

## Effect of TFIID disassembly on transcription

Given the essential role of TAF4 in TFIID core assembly, the lack of TAF4b to compensate for its absence, and the reduced TBP-TAF co-precipitation, it is likely that TFIID integrity is disrupted in mutant hepatocytes, with the residual TBP-TAF co-precipitation reflecting the presence of intact TFIID in the cell types where TAF4 remains expressed. A similar conclusion was drawn when TAF10 that is also essential for TFIID integrity was inactivated in adult liver (*Tatarakis et al., 2008*). TAF10 inactivation occurred later than TAF4 in this study and despite TFIID disassembly, a milder phenotype is observed as *Taf10*hep−/− animals die at P35–P38. Upon TAF10 inactivation, recruitment of TBP and TAFs to many promoters is lost but PIC formation was not affected suggesting that transcription persists in the absence of TBP/TFIID. In contrast, we show that TBP (and TAF1, unpublished data) is recruited to promoters of expressed genes, and its recruitment and PIC formation are diminished to comparable extents. Moreover, TAF3 promoter occupancy at expressed genes is less affected. While loss of TFIID integrity compromises TBP occupancy and PIC formation, interaction with H3K4me3 acts to stabilise TAF3 recruitment independently of TBP. We therefore find no evidence for PIC formation and gene expression in the absence of TBP and TAF3 recruitment in the TAF4 mutant hepatocytes.

TAF4 regulates hepatocyte gene expression by controlling HNF4A occupancy, PIC formation, and Pol II pausing. Expression of subset of embryonic genes is up-regulated in the absence of TAF4 due to decreased Pol II pausing, although it is also possible that changes in RNA stability may also contribute to the observed increase in mRNA abundance. Why TAF4 regulates pausing at this set of genes remains to be determined, although we note that they are enriched in short intron-less genes. Also it has been previously described that TAF7 has a negative effect on CDK9 and BRD4 activity (*Devaiah et al., 2010*). Perhaps defective recruitment of TAF7 at these genes due to compromised TFIID integrity would explain the augmented CDK9 recruitment and an increase in its kinase activity would account for the enhanced elongation. Nevertheless, our observations show that down-regulation of many of these genes normally occurs at least initially through increased Pol II-pausing at post-natal stages, while in the absence of TAF4 they continue to be expressed with higher levels of elongating Pol II as in their embryonic mode. We also observed diminished paused Pol II at a small subgroup of highly expressed liver identity genes that may be under the control of 'super enhancers' (*Whyte et al.,*

*2013*), where TAF3 and H3K4me3 levels were also reduced. This concomitant reduction suggests that TAF3/H3K4me3 levels may modulate Pol II pausing at this class of genes.

## Conclusion

We show that TAF4 is essential for activation of the post-natal hepatocyte gene expression programme acting as a cofactor for HNF4A. Previous models proposed that activators bind their cognate sites and promote PIC formation via interactions with TAFs. We show in vivo that HNF4A stably occupies a set of functional sites only when accompanied by concomitant TAF4-dependent PIC formation showing that these are mutually dependent events (*Figure 9D*). Many recent studies have highlighted the role of long-range enhancer–promoter interactions and the formation of large-scale chromatin domains (*de Laat and Duboule, 2013*). In contrast, we highlight here the importance of local interactions at the TSS. Although HNF4A is also present at distal enhancers, functional HNF4A sites cluster close to the TSS and interactions with TFIID and also possibly the mediator complex (*Malik et al., 2002*) play a critical role in gene activation.

## Materials and methods

### Animals

The previously described (*Mengus et al., 2005*) *Taf4a*^lox/lox animals were crossed with Albumin-Cre (*Alb*-Cre) transgenic mice (*Postic and Magnuson, 2000*). Experiments were performed in compliance with National Animal Care Guidelines (European Commission directive 86/609/CEE; French decree no. 87–848).

### mRNA-seq and ChIP-seq

Library preparation and mRNA sequencing were performed as previously described (*Herquel et al., 2013*). The results were confirmed by qRT-PCR for at least 30 independent genes. Gene functional annotation was performed using DAVID (http://david.abcc.ncifcrf.gov/).

For ChIP-seq, mouse livers, freshly isolated or snap frozen in liquid nitrogen, were homogenized by douncing and fixed with 1% PFA for 10 min and fixing was stopped by adding glycine at a final concentration of 0.125 M. Alternatively, for TAF3 and for TBP ChIP-seq, chromatin was fragmented by MNase I digestion as follows. Fixed nuclei were resuspended in equal volume of MNase buffer (50 mM Tris–HCl, pH 8.0, 15 mM NaCl, 5 mM CaCl$_2$, 60 mM KCl) and treated with MNase (#MO247S; NEB Ipswich, MA) at 37°C for 10 min. The reaction was stopped by addition of EDTA to a final concentration of 20 mM. Chromatin was incubated with 0.3% SDS (5 min on ice) and with 1% Triton X-100 (5 min on ice) and centrifuged at 14,000 rpm. ChIP was performed overnight with 50 µg of chromatin and 5 µg of the following antibodies: anti-Pol II (sc-9001 X), anti-TFIIB (sc-225 X), anti-TFIIE (sc-6935 X), anti-HNF4A (sc-8987 X), anti-CDK9 (sc-8338 X), anti-HNF6 (sc-13050 X), anti-CEBPA (sc-9314 X), anti-FOXA2 (sc-6554 X) from Santa Cruz (Santa Cruz, CA), anti-TBP (ab28175) from Abcam (Cambridge UK), anti-CTCF (07–729) and anti-H3K4me3 (04–745) from Millipore (Billerica, MA), anti-TAF3 (IGBMC, in house). ChIP-seq libraries were prepared as previously described and sequenced on the Illumina Hi-seq2500 as single-end 36-base reads (*Choukrallah et al., 2012*). Peak detection was performed using the MACS software (*Zhang et al., 2008*) using the no antibody ChIP as negative control. Data sets were normalised for the number of unique mapped reads for subsequent comparisons. Global clustering analysis and quantitative comparisons were performed using seqMINER (http://bips.u-strasbg.fr/seqminer/) (*Ye et al., 2011*) and R (http://www.r-project.org/) and visualized using *ggplot2* package. Genomic region annotation was performed with either seqMINER or GREAT (http://bejerano.stanford.edu/great/public/html/). Overlap between HNF4A-binding sites that are enhanced or depleted in TAF4-mutant liver and CRMs was performed by intersecting the corresponding genomic coordinates.

### Extract preparation, immunoprecipitation, and Western blot

Liver nuclei extracts were made as follows. Livers were homogenized by douncing in cold hypotonic buffer with 25 mM HEPES, pH 7.8, 1.5 mM MgCl$_2$, 10 mM KCl, and 0.1% NP-40, supplemented with Protease Inhibitor Cocktail (Roche, Basel, Switzerland) and DTT. Nuclei were washed twice in cold hypotonic buffer followed by centrifugation (3000 rpm, 5 min) and lysed in 50 mM Tris–HCl, pH 7.5, 450 mM NaCl, 0.5% NP40, 5% glycerol for 2 hr on 4°C with constant rotation. Detection was performed using the same antibodies as for ChIP or previously described in house antibodies against

TBP and TAFs (*Gangloff et al., 2001*). TAFs, HNF4A, CDK9, Pol II, and TBP were revealed with the antibodies described elsewhere in the 'Materials and methods'. Immunoprecipitation was performed overnight with anti-TBP (3G3) or anti-HA as control.

## Histochemistry and immunofluorescence

Livers were fixed at 4°C for 16 hr in 4% paraformaldehyde in phosphate-buffered saline (PBS), washed overnight in PBS, and embedded in paraffin. For immunofluorescence 10 µm sections were treated in a microwave oven in citrate buffer pH 6.0 for 15 min at 150 W. Staining was performed with primary anti-TAF4 (sc-136093) and anti-HNF4A (sc-6556) from Santa Cruz, Claudin3 (ab15102) and TBP (ab51841) from Abcam, Sox9 (AB5535) from Chemicon, anti-aSMA (M0851) from DAKO, TJP1 from Invitrogen (Carlsbad, CA) (339100), KI67 from Novocastra (Nussloch, Germany) (NCL-KI67P). Nuclei were counterstained with Hoechst. For histological analysis, sections were stained with Hematoxylin and Eosin, Red Oil O, and *Periodic* acid–*Schiff* (PAS) following the standard procedures.

## Bacterial co-expression of TAF4, TAF12, and HNF4 and complex purification

Full length HNF4A and its indicated deletion mutants, the histone-fold containing fragments of human TAF4 (805–1083) and (832–966) and TAF12 (29–161), were cloned into the His-Tag vector pEA-tH and/or the native pCS vector (*Diebold et al., 2011*). *E. coli* BL21 cells were transformed with respective vectors and colonies were used to grow 15 ml cultures at 37°C till the OD600 = 0.4. Protein expression was induced by IPTG and cultures were grown at 25°C overnight. Cells were resuspended in 1.5 ml of lysis buffer (10 mM Tris–HCl, pH 8.0 and 50 mM, 200 mM or 400 mM NaCl) and sonicated. After the centrifugation, the supernatants were incubated with TALON resin for two hours at 4°C with the permanent agitation. The resin was washed twice with 1 ml of ice-cold lysis buffer and resuspended in 25 µl of Laemmli buffer. Samples were analysed by SDS-PAGE with subsequent Coomassie brilliant blue staining.

## RNA extraction and quantitative real-time RT-PCR

Total RNA was extracted with the RNeasy Mini Kit (Qiagen, Venlo, Holland) following the manufacturer's instructions and treated with RNase-free DNase (Fermentas). RNA (3 µg) was reverse transcribed by using a Maxima Kit (Fermantas, Pitsburgh, PA). The final product was then diluted 10 times and 2 µl were mixed with forward and reverse primers (250 nM of each primer at final concentration) and 5 µl of SYBR Green master mix (Qiagen). The real-time PCR reaction was performed by using the LightCycler 1.5 system (Roche). Each cDNA sample was tested at least in triplicate.

## Acknowledgements

We thank I Michel for excellent technical assistance, Dr N Engelhardt and Dr N Lazarevich for monoclonal antibodies and HNF4A expression vector respectively; Dr P Jacquemin for advice and comments, all staff of the IGBMC and ICS common services in particular the sequencing and bioinformatics platform. This work was supported by institutional grants from the Centre National de la Recherche Scientifique, the Institut National de Sante et de la Recherche Médicale, the Université de Strasbourg, the Ligue Nationale contre le Cancer, the ANR-10-LABX-0030-INRT frame programme. ID is an 'équipe labellisée' of the Ligue Nationale contre le Cancer. The IGBMC high throughput sequencing facility is a member of the 'France Génomique' consortium (ANR10-INBS-09-08). DA was supported by the fellowship from La Fondation pour la Recherche Médicale.

The data described here have been deposited in the GEO data base under the access code GSE57814 and the data in *Supplementary file 1* are deposited in Dryad, with reference doi:10.5061/dryad.62gj0.

---

## Additional information

### Funding

| Funder | Author |
| --- | --- |
| Centre National de la Recherche Scientifique | Irwin Davidson |
| Institut National de la Santé et de la Recherche Médicale | Irwin Davidson |

| Funder | Author |
|---|---|
| Ligue national contre le cancer | Irwin Davidson |
| Agence Nationale de la Recherche | Irwin Davidson |

The funders had no role in study design, data collection and interpretation, or the decision to submit the work for publication.

## Author contributions
DA, Designed, performed and analysed 1); histopathology analysis of the mutant mice, 2) immunostainings, 3) the ChIP-seq experiments, Performed bioinformatics analyses and wrote the article together with corresponding author; DL, Designed, performed and analysed the immunoprecipitation experiments; BB, Provided data on CRM locations and performed bioinformatics analysis in collaboration with first author; SLG, Performed bioinformatic analysis in collaboration with first author; CR, Performed and analysed bacterial co-expression and protein purification; GM, Responsible for mouse breeding, genotyping, dissection and histology; ID, Conceived and designed the project, Analysed data, Wrote the manuscript together with first author, Overall project management

## Ethics
Animal experimentation: Animal Experiments were performed in compliance with National Animal Care Guidelines (European Commission directive 86/609/CEE; French decree no.87-848).

# Additional files

## Supplementary file
• Supplementary file 1. Excel table of RNA-seq data. Pages 1 and 2 show the genes down- and up-regulated in TAF4 mutant liver respectively. Page 3 indicates genes with associated HNF4A occupied sites in E18.5 liver and page 4 the ontology terms associated with these genes. Page 5 lists genes that show loss of HNF4A occupancy and down-regulation in TAF4 mutant liver and page 6 the ontology terms associated with these genes (*Supplementary file 1*).

## Major datasets

The following datasets were generated:

| Author(s) | Year | Dataset title | Dataset ID and/or URL | Database, license, and accessibility information |
|---|---|---|---|---|
| Alpern D, Langer D, Ballester B, Le Gras S, Romier C, Mengus G, Davidson I | 2014 | TAF4 promotes pre-initiation complex formation and HNF4A occupancy of regulatory elements required to activation post-natal gene expression programme in hepatocytes GSE57814 | http://www.ncbi.nlm.nih.gov/geo/query/acc.cgi?acc=GSE57814 | Publicly available at NCBI GEO. |
| Alpern D, Langer D, Ballester B, Le Gras S, Romier C, Mengus G, Davidson I | 2014 | Gene expression changes upon inactivation of TFIID subunit TAF4 in neonatal mouse liver | doi:10.5061/dryad.62gj0 | Dryad. |

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
