## [Decision Letter]

Thank you for sending your work entitled “TAF4 directs HNF4A promoter occupancy to induce post-natal hepatocyte differentiation” for consideration at *eLife*. Your article has been favorably evaluated by Stylianos Antonarakis (Senior editor) and 3 reviewers, one of whom is a member of our Board of Reviewing Editors.

The following individuals responsible for the peer review of your submission have agreed to reveal their identity: Michael R Green (Reviewing editor), Robert Roeder (one of the two other reviewers).

The Reviewing editor and the other reviewers discussed their comments before we reached this decision, and the Reviewing editor has assembled the following comments to help you prepare a revised submission.

This manuscript elaborates a specific TAF4 requirement for liver development and functions that reflects a major contribution to the HNF4a-mediated genomic program. By using a liver-specific TAF4 mutant mouse line, the authors observed several severe liver failures in P12 to P21 mice that led to lethality. RNA-seq analysis suggest that a subset of liver functional genes are deregulated in TAF4 mutant mice. Using genome-wide ChIP-seq assays, the authors further show that TAF4 inactivation diminishes PIC formation and attenuates HNF4a occupancy on a subset of post-natally expressed genes in liver. Finally, in a biochemical analysis the authors show a direct interaction between HNF4a and the TAF4/TAF12 dimer that may explain the impaired HNF4a occupancy of its targets in TAF4 mutant mice.

Overall, the results presented in this manuscript provide substantial evidence for the physiological functions of a TAF subunit that is specifically required for HNF4a-controlled gene expression related to liver function. Although previous studies have shown a TAF10 subunit requirement for liver function and TAF coactivator functions in cell-based assays, the current manuscript provides significant new evidence for physiological TAF coactivator functions by showing, first, a TAF4 function in the genome-wide occupancy of a TAF4-interacting activator and, second, a mechanism involving direct interactions.

However, the three reviewers have a number of specific concerns and the authors are encouraged to address as many of these as possible with experimentation and/or textual revisions as appropriate.

Major specific comments:

1) The main conclusion that Taf4 is a cofactor for Hnf4a during liver maturation is not supported by the data provided in this paper. Although the authors show that the Taf4-Taf12 heterodimer interacts with Hnf4a when the recombinant proteins are expressed in E. coli, these data do not support a physiological interaction of these two proteins during liver maturation. Additionally, due to the fact that Hnf4a has been previously shown to directly interact with TBP (38), and the data in this paper show that TBP does not interact with the TFIID complex properly when Taf4 is conditionally knocked out, and TBP is obsolete from the majority of its genomic binding locations in the Taf4 depleted livers, it seems appropriate for the authors to conclude that the mis-localization of Hnf4a at its genomic binding sites in the Taf4 depleted livers results from a secondary effect (improper TFIID assembly which results in a loss of TBP at Hnf4a target genes).

2) Another explanation for the altered genomic binding locations of Hnf4a in Taf4 depleted livers may be that the loss of Taf4 results in altered genomic locations of other Hnf4a cofactors. Several other cofactors have been identified for Hnf4a, including CBP/p300. It would add to the breadth of the paper, and allow for a more succinct conclusion to be made about the disrupted genomic binding of Hnf4a in the Taf4-/- livers, if analysis was performed to determine whether or not CBP/p300 or other Hnf4a cofactors are lost at Hnf4a binding locations when Taf4 is conditionally ablated.

3) In Figure 7, the authors observed a reduction of TSS-bound Pol II (paused) in the up-regulated gene subset. The explanation offered is that TAF4 inactivation increases the recruitment of CDK9 (Figure 7), leading in turn to release of paused Pol II and increased gene expression. However, it also is noted that PIC formation is diminished on those genes (Figure 6). Could the authors provide a possible mechanism for reduced PIC formation leading to enhance CDK9 recruitment and Pol II release?

4) In Figure 7, the authors show increased Pol II occupancy for a few up-regulated genes (Figure 7). However the Pol II metaprofile (Figure 7) does not show a significant enhancement for up-regulated genes. Since the RNA-seq measures accumulated mRNA molecules, other mechanisms, such as mRNA stability, should be discussed.

5) In Figure 6, it is noted that TAF4 inactivation globally diminishes the occupancies of TBP, TFIIB, TFIIE and H3K4Me3, but has less effect on TAF3 binding. Is this observed variation in TAF3 binding due to an alternative chromatin fragmentation method used for TAF3 ChIP or is there something unique about TAF3 relative to TFIID-specific TAFs? Was any other TAF analyzed?

6) In Figure 9, an analysis of HNF4a-enriched sites should be added as well.

7) The authors should discuss how a GTF (TAF4) recruits a sequence-specific DNA binding protein (HNF4A). The more general model is that the sequence-specific DNA binding protein recruits the GTF. Likewise, the authors should discuss what might be the basis of occupancy of HNF4A at new sites in the absence of TAF4.

8) In Figure 9, what is the criterion used (e.g., distance from TSS) for association between HNF4A and a gene?

9) In Figure 7 the authors need to explain in the Results section how paused pol II downstream of the transcription start-site is distinguished from the pol II of the PIC.

10) The authors conclude that down-regulation is due to defective PIC formation, while up-regulation of many genes reflects decreased pausing and increased elongating Pol II and CDK9 recruitment. Do the authors have any ideas as to how TAF4 can act in such a dual fashion and the basis by which TAF4 behaves so differently at down and up regulated genes?

11) Figure 7. Have the authors performed a similar experiment measuring CDK9 recruitment for up-regulated genes?

12) The TAF3 results, although interesting, don't really connect well with the major conclusions of this study, which are TAF4, HNF4A and liver development. This reviewer at least found these to be a distraction. The authors might consider removing these experiments or at least de-emphasizing them by for example moving them to supplemental.

---

## [Author Response]

*1) The main conclusion that Taf4 is a cofactor for Hnf4a during liver maturation is not supported by the data provided in this paper. Although the authors show that the Taf4-Taf12 heterodimer interacts with Hnf4a when the recombinant proteins are expressed in E. coli, these data do not support a physiological interaction of these two proteins during liver maturation. Additionally, due to the fact that Hnf4a has been previously shown to directly interact with TBP (*[38]*), and the data in this paper show that TBP does not interact with the TFIID complex properly when Taf4 is conditionally knocked out, and TBP is obsolete from the majority of its genomic binding locations in the Taf4 depleted livers, it seems appropriate for the authors to conclude that the mis-localization of Hnf4a at its genomic binding sites in the Taf4 depleted livers results from a secondary effect (improper TFIID assembly which results in a loss of TBP at Hnf4a target genes)*.

The referees raise several points here and we have included some additional data that reinforce the idea that HNF4A-TAF4-TAF12 interactions are also likely to contribute to HNF4A localisation and PIC formation.

Firstly, we agree with the referees that the HNF4-TBP interaction may be involved in HNF4A localisation and PIC formation. This is why we illustrated this in our summary model (Figure 9 in the original version of the paper). Also in the Discussion section we stated ‘While this may in part be accounted for by HNF4A-TBP interactions, we show that HNF4A also interacts with the TAF4-TAF12 heterodimer.’ We also stated ‘…strongly suggesting that direct HNF4A-TFIID interactions via the TAF4-TAF12 heterodimer and perhaps also TBP are required to activate liver function genes.’ Thus, we did not want to imply that the HNF4A-TAF4-TAF12 interaction was the only one involved. We have modified the text of the revised Discussion section to try and make this clearer.

Furthermore, we now show in new data added in the new Figure 9 of the revised version that TAF4-TAF12 interacts with the LBD of HNF4A. This is complementary to the TBP-DBD interaction reported by Takahashi et al. These new data indicate that HNF4A can make contacts with TFIID both through its DBD and its LBD. Indeed, as the referees point out the defective TFIID assembly in the mutant suggests that interactions with TBP may contribute to, but are not sufficient for HNF4A recruitment and PIC assembly and that multiple interactions with TFIID through TBP and the associated TAF4-TAF12 are necessary. The new data have been added to the text and the Discussion section has been modified. We believe that this new data strengthens our original findings of HNF4A-TAF4-TAF12 interactions.

*2) Another explanation for the altered genomic binding locations of Hnf4a in Taf4 depleted livers may be that the loss of Taf4 results in altered genomic locations of other Hnf4a cofactors. Several other cofactors have been identified for Hnf4a, including CBP/p300. It would add to the breadth of the paper, and allow for a more succinct conclusion to be made about the disrupted genomic binding of Hnf4a in the Taf4-/- livers, if analysis was performed to determine whether or not CBP/p300 or other Hnf4a cofactors are lost at Hnf4a binding locations when Taf4 is conditionally ablated*.

Based on the suggestion of the referees, we have attempted to perform anti-p300 ChIP-qPCR experiments, but we saw little enrichment even at HNF4A-occupied sites where public ChIPseq data shows that p300 should be present. While this is probably a technical issue that would take some time to resolve, we would like to stress however that in the original manuscript, we showed that Cebpa, Foxa2 and Hnf6 were all depleted along with Hnf4a at CRMs (Figure 9–figure supplement 1). As it is the activators that recruit the co-activators rather than the co-activators that recruit the activators, it would seem highly unlikely that changes in p300 recruitment are the principal cause of the lost HNF4Aa binding. In addition, our new data showing that TAF4-TAF12 interacts with the HNF4a LBD reinforce the idea that a specific interaction with HNF4A is the important one.

*3) In*
Figure 7*, the authors observed a reduction of TSS-bound Pol II (paused) in the up-regulated gene subset. The explanation offered is that TAF4 inactivation increases the recruitment of CDK9 (*Figure 7*), leading in turn to release of paused Pol II and increased gene expression. However, it also is noted that PIC formation is diminished on those genes (*Figure 6*). Could the authors provide a possible mechanism for reduced PIC formation leading to enhance CDK9 recruitment and Pol II release?*

For the moment, we do not have a specific mechanism that would explain this. As it is likely that TFIID integrity is compromised, perhaps the presence of TAF4 or other TAFs at these promoters that is lost in the mutant, has a negative effect on CDK9 recruitment. It has been previously described by the Singer lab that TAF7 has a negative effect on CDK9 and BRD4 activity. Perhaps defective recruitment of TAF7 at these genes due to compromised TFIID integrity would explain the augmented CDK9 recruitment and an increase in its kinase activity would account for the enhanced elongation. We did not discuss this as we have no ChIP-grade TAF7 antibody that allows us to test the idea. Also enhanced elongation is not a general effect, but is limited to a sub-set of genes. We originally investigated whether these promoters had some common feature (TATA+, or TATA-less for example), but so far we have not identified any common feature of this subset of promoters. We have included a discussion of this in the revised version.

*4) In*
Figure 7*, the authors show increased Pol II occupancy for a few up-regulated genes (*Figure 7)*. However the Pol II metaprofile (*Figure 7*) does not show a significant enhancement for up-regulated genes. Since the RNA-seq measures accumulated mRNA molecules, other mechanisms, such as mRNA stability, should be discussed*.

We fully agree with the referees and we have added a word factor in the revised version on the possibility of changes in mRNA stability as a contributing factor.

*5) In*
Figure 6*, it is noted that TAF4 inactivation globally diminishes the occupancies of TBP, TFIIB, TFIIE and H3K4Me3, but has less effect on TAF3 binding. Is this observed variation in TAF3 binding due to an alternative chromatin fragmentation method used for TAF3 ChIP or is there something unique about TAF3 relative to TFIID-specific TAFs? Was any other TAF analyzed?*

In fact TAF3 was the not only protein that we analysed by MNaseI digestion. TBP was also analysed in this way. We have corrected the Materials and methods section to clarify this point as it was not correctly described in the original version. The classic sonication method does not work with our anti-TAF3 antibody and this is why MNaseI was used, while for TBP, ChIP is much more efficient with MNaseI than sonication. Thus, it is not the chromatin fragmentation method that accounts for the lesser effect seen with TAF3. We rather propose that the high affinity of TAF3 for H3K4me3, a unique feature of TAF3 relative to other TAFs, accounts for the observed data.

*6) In*
Figure 9*, an analysis of HNF4a-enriched sites should be added as well*.

We have added this additional analysis requested by the referees (Figure 8 in the new version of the paper). In our new analysis, we show that these sites have no specific location, in particular no striking enrichment at the TSS as seen with depleted sites, although a small number are associated with up-regulated genes. This small set perhaps corresponds to sites that are occupied during embryogenesis and normally lost in neonatal liver, but not in the mutant. The vast majority of sites are however not in this category. It is more likely that there is a more general disorganisation of the chromatin landscape in the mutant liver and that this allows accessibility to sites that are not normally accessible to HNF4A. The text has been modified accordingly.

*7) The authors should discuss how a GTF (TAF4) recruits a sequence-specific DNA binding protein (HNF4A). The more general model is that the sequence-specific DNA binding protein recruits the GTF. Likewise, the authors should discuss what might be the basis of occupancy of HNF4A at new sites in the absence of TAF4*.

In the Discussion section of the paper, we tried to emphasize that PIC formation and HNF4A binding are mutually dependent events. Our data are consistent with a model where stable binding of HNF4A to TSS proximal CRMs requires interaction with TFIID and PIC formation. Mutual interactions between HNF4A and TFIID stabilise both PIC formation and CRM occupancy, without PIC formation HNF4A does not stably bind to its binding sites. In our data we showed that a majority of sites occupied in absence of TAF4 are not normally occupied either during embryogenesis or in adult liver. As mentioned above, sites enriched in the TAF4 mutant have no specific location although a small number are associated with upregulated genes. This small set perhaps corresponds to sites that are occupied during embryogenesis and normally lost in neonatal liver, but not in the mutant. The vast majority of sites are however not in this category. It is more likely that there is a more general disorganisation of the chromatin landscape in the mutant liver and that this allows accessibility to sites that are not accessible to HNF4A.

*8) In*
Figure 9*, what is the criterion used (e.g., distance from TSS) for association between HNF4A and a gene?*

In Figure 9 (new Figure 8) we used a window of +/-40 kb with respect to the TSS. This was indicated in the figure and in the figure legend, but not in the text. This has been corrected.

*9) In*
Figure 7
*the authors need to explain in the Results section how paused pol II downstream of the transcription start-site is distinguished from the pol II of the PIC*.

We do not understand exactly what the referees are referring to here. It is now well established that ‘paused’ Pol II accumulates around 50 nucleotides downstream of the TSS at a large number of promoters. This is what is visualized in the ChIP-seq experiments.

10) The authors conclude that down-regulation is due to defective PIC formation, while up-regulation of many genes reflects decreased pausing and increased elongating Pol II and CDK9 recruitment. Do the authors have any ideas as to how TAF4 can act in such a dual fashion and the basis by which TAF4 behaves so differently at down and up regulated genes?

This is related to comments 3 and 7. Please see discussion in reply to these comments.

*11)*
Figure 7*. Have the authors performed a similar experiment measuring CDK9 recruitment for up-regulated genes?*

I presume that the referees are referring to ‘down-regulated’ genes, as up-regulated genes were already shown in the original manuscript. Yes, we had performed these experiments and there is no enhanced CDK9 recruitment at down-regulated genes. We have now included one example in Figure 7 to illustrate this.

*12) The TAF3 results, although interesting, don't really connect well with the major conclusions of this study, which are TAF4, HNF4A and liver development. This reviewer at least found these to be a distraction. The authors might consider removing these experiments or at least de-emphasizing them by for example moving them to supplemental*.

We fully agree with the comment of the referees: the TAF3 data are interesting but not central to the story. We have therefore moved these data into the supplemental section, new Figure 6—figure supplement 1, thus allowing addition of the new co-expression data that has been moved to a new Figure 9, without increasing the overall number of figures.